# Self-nested large-eddy simulations in PALM Model System v21.10 for offshore wind prediction under different atmospheric stability conditions

Maria Krutova[1], Mostafa Bakhoday-Paskyabi[1], Joachim Reuder[1], and Finn Gunnar Nielsen[1]

[1]Geophysical institute and Bergen Offshore Wind Centre, University of Bergen, Allégaten 70, 5007 Bergen, Norway

**Correspondence:** Maria Krutova (maria.krutova@uib.no), Mostafa Bakhoday-Paskyabi (mostafa.bakhoday-paskyabi@uib.no)

**Abstract.**

Large-eddy simulation (LES) resolves large-scale turbulence directly and parametrizes small-scale turbulence. Resolving micro-scale turbulence, e.g., in wind turbine wakes, requires both a sufficiently small grid spacing and a domain large enough to develop turbulent flow. Refining a grid locally via a nesting interface effectively decreases the required computational time compared to the global grid refinement. However, interpolating the flow between nested grid boundaries introduces another source of uncertainty. Previous studies reviewed nesting effects for a buoyancy-driven flow and observed a secondary circulation in the two-way nested area. Using a nesting interface with a shear-driven flow in LES, therefore, requires additional verification. We use PALM Model System 21.10 to simulate a boundary layer in a cascading self-nested domain under neutral, convective, and stable conditions and verify the results based on the wind speed measurements taken at the FINO1 platform in the North Sea. We show that the feedback between parent and child domains in a two-way nested simulation of a non-neutral boundary layer alters the circulation in the nested area, despite spectral characteristics following the reference measurements. Unlike the pure buoyancy-driven flow, a non-neutral shear-driven flow slows down in a two-way nested area and accelerates after exiting the child domain. We also briefly review the nesting effect on the velocity profiles and turbulence anisotropy.

## 1 Introduction

Large-eddy simulation (LES) allows performing a detailed process study for areas and situations where we lack appropriate field measurements. For this reason, LES is widely used for high-fidelity simulations of wind flows in wind energy applications. When considering the turbulent flow, the grid resolution should be sufficiently high to resolve the relevant turbulence scales (Wurps et al., 2020). Increased grid resolution comes at the cost of gradually increased computational time. The overall computational time can be reduced by refining a grid locally through the nesting interface. While improving the grid resolution, a nesting interface introduces new uncertainties in the simulation. Such nesting effects are documented for buoyancy-driven flows, with the strongest influence observed for the two-way nesting mode (Moeng et al., 2007; Hellsten et al., 2021). A buoyancy-driven flow develops a secondary circulation and decreased velocity inside the nested area – the effect becomes prominent for the data averaged over several hours. However, buoyancy-driven flows are characterized by near-zero wind

**Table 1.** Aggregated statistics of 1-hour sonic anemometer time series.

| Stability | $\overline{U}_{119}$, ms$^{-1}$ | $TI_{80}$, % | $L$, m | $\zeta$ | $\psi$ | 1-hour period start |
|-----------|----------|----------|--------|---------|--------|---------------------|
| NBL | 12.41 | 6.6 | 2753 | 0.043 | 0 | April 18, 2016 04:30 |
| CBL | 12.58 | 6.1 | -451 | -0.263 | 0.528 | February 22, 2016 19:00 |
| SBL | 12.14 | 3.2 | 158 | 0.753 | -3.540 | June 2, 2016 16:30 |

speed, while the wind energy research primarily deals with wind speeds of $5 - 25\,\text{ms}^{-1}$. Therefore, shear-driven LES with the nesting interface require additional verification.

We use a Fortran-based LES code PALM 21.10 (Maronga et al., 2020) to simulate wind flow with a speed of $12.5\,\text{ms}^{-1}$ at the reference height of 119 m for three stability conditions: true neutral (NBL), convective (CBL), and stable (SBL) boundary layers. The initial velocity and turbulence intensity profiles are defined to match 1-hour averages of the sonic anemometer time series as processed by Nybø et al. (2019). The domain is simulated for a non-nested grid and nested grids with one-way or two-way nesting modes. The resulting turbulence statistics are then compared with the measurements to evaluate the model's performance.

## 2 Data

The reference measurements contain wind speed directional components $u$, $v$, and $w$ recorded with sonic anemometers during the Offshore Boundary-Layer Experiment at FINO1 (OBLEX-F1) campaign in $2015-2016$ in the North Sea. The meteorological mast is installed on the FINO1 platform located in the North Sea at $54°\,00'\,53.5''$N, $6°\,35'\,15.5''$E, 45 km to the north of the German island of Borkum.

The sonic anemometers were installed at the meteorological mast at 40, 60, and 80 m. The measurements were processed by Nybø et al. (2019) and organized into one-hour time series of 1 Hz frequency. Each processed series corresponds to different pairs of a stability condition and mean wind speed at the reference height of 119 m. This height was chosen as an outlook into future wind turbine development and corresponds to a hub height of the DTU reference 10 MW turbine (Bak et al., 2013). The reference height unifies different stability conditions under the assumption of a similar flow speed. Due to the computational time restrictions, we simulate only those conditions where the horizontal wind speed reaches approximately $\overline{U}_{119} = 12.5\,\text{ms}^{-1}$ at the reference height (Table 1).

The wind speed $\overline{U}_{119}$ at the reference height was estimated from the measurement data. Since the measurements are originally available only for three levels, the mean wind speed profile was approximated by Nybø et al. (2020) by fitting the logarithmic law

$$\overline{u}(z) = u_{ref} \left[ \frac{\ln\left(\frac{z}{z_0} - \psi\right)}{\ln\left(\frac{z_{ref}}{z_0} - \psi\right)} \right] \tag{1}$$

where the reference wind speed $u_{ref}$ is taken for the reference height $z_{ref} = 80\,\mathrm{m}$, and the stability correction function $\psi$ is defined as in (Stull, 1988)

$$\psi = \begin{cases} 0 & -\text{ NBL}, \\ -2\ln\dfrac{1+x}{2} - \ln\dfrac{1+x^2}{2} + 2\arctan x - \dfrac{\pi}{2} & -\text{ CBL}, \\ 4.7\zeta & -\text{ SBL}, \end{cases} \tag{2}$$

where $x = (1 - 15\zeta)^{1/4}$. The stability parameter $\zeta$ is derived from the height above the surface $z$ and Obukhov length $L$ as

$$\zeta = \frac{z}{L} \tag{3}$$

The roughness length $z_0$ in Eq. (1) is, therefore, a fitting parameter to be found. The estimation is performed under an assumption of a boundary layer starting above $119\,\mathrm{m}$ and is applicable only to the mean wind profile. During the simulation, we attempt to match the mean wind profile, including the estimated wind speed at $119\,\mathrm{m}$ and turbulence intensity calculated for levels 40, 60, and $80\,\mathrm{m}$.

## 3 Methodology

### 3.1 PALM LES model

We perform a free-flow large-eddy simulation (LES) using the Fortran code PALM developed at Universität Hannover (Maronga et al., 2020). PALM utilizes a staggered Arakawa C-grid: the velocity components are defined at the grid cell edges and are shifted by a half grid spacing; the scalar variables are defined at the center of a grid cell. The subgrid-scale fluxes are resolved via the Deardorff 1.5-order closure model.

By default, PALM solves prognostic equations for the velocity components $u$, $v$, $w$, and potential temperature $\theta$. If the stability condition is set to true neutral, the temperature is considered constant, and the corresponding equation is not solved. Buoyancy terms are also not considered in a true neutral simulation

A nested simulation in PALM consists of at least one child domain inside a parent domain. Each child domain can simultaneously be a parent domain for another child domain, thus forming a cascading self-nested structure. The top-level parent domain is further referred to as the root domain to make a distinction from inner parent domains. Overall, PALM supports simulation of one root domain and up to 63 child domains.

The nesting algorithm is constructed in a way to optimize computational time for multiple child domains (Hellsten et al., 2021). The nested domains communicate via interpolation which is performed just before the pressure-correction step, so that the time-consuming pressure solver is run only once per the time step. The solution at the nested boundaries of a parent domain – velocity components and scalar quantities, e.g., temperature and humidity – is linearly interpolated to all nested boundaries, except the bottom surface, as boundary conditions. The bottom surface is always located at a zero level as in the root domain and utilizes Dirichlet or Neumann boundary conditions as prescribed in the corresponding child domain input files.

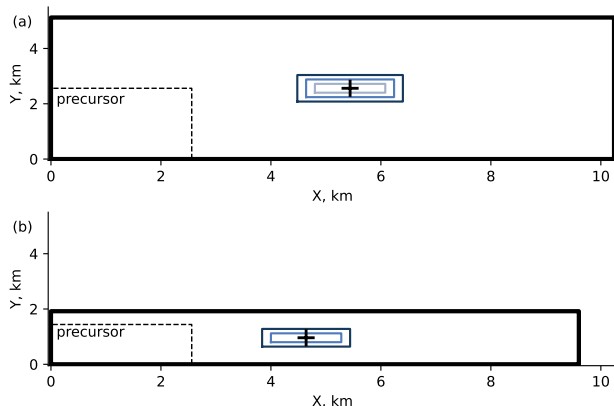

**Figure 1.** Nested domains schematic. (a) NBL and CBL domains, (b) SBL domains.

**Table 2.** Grid parameters for NBL and CBL nested domains (Fig. 1a).

| Domain | $N_x$ | $N_y$ | $N_z$ | $\Delta_x$, m | Bottom-left corner $x$, m | $y$, m |
|---|---|---|---|---|---|---|
| Precursor | 256 | 256 | 160 | 10 | - | - |
| Precursor | 512 | 512 | 256 | 5 | - | - |
| Root | 1024 | 512 | 160 | 10 | - | - |
| Child #1 | 384 | 192 | 128 | 5 | 4480 | 2080 |
| Child #2 | 640 | 256 | 192 | 2.5 | 4640 | 2240 |
| Child #3 | 1024 | 256 | 256 | 1.25 | 4800 | 2400 |

After the interpolation, the prognostic equations are solved for a child domain. In the case of cascading nesting, the procedure is repeated until the solution is found for all nested domains at the current step. In a one-way nesting case, the simulation proceeds to the pressure-correction step, so the solution in parent domains remains unaffected by the solution in child domains. In a two-way nesting case, PALM uses an anterpolation scheme proposed by Clark and Farley (1984) to return a child domain's

solution to the parent domain; technical details behind the implementation are explained in Hellsten et al. (2021). Each child domain anterpolates its solution via first-order integration to the respective parent domain before the pressure-correction step. Therefore, the two-way nested solution remains similar in the nested area, while the one-way nested solution may eventually diverge for parent and child domains.

### 3.2 Precursor and main LES run parameters

One of the ways PALM can simulate a turbulent flow is a precursor-main run scheme, which does not require complex dynamic input data and effectively reduces the domain size required for turbulence development (Witha et al., 2014). First, a small

**Table 3.** Grid parameters for SBL nested domains (Fig. 1b).

| Domain | $N_x$ | $N_y$ | $N_z$ | $\Delta_x$, m | Bottom-left corner $x$, m | $y$, m |
|--------|-------|-------|-------|---------------|----------|--------|
| Precursor | 512 | 288 | 160 | 5 | - | - |
| Root | 1920 | 384 | 160 | 5 | - | - |
| Child #1 | 640 | 256 | 192 | 2.5 | 3840 | 640 |
| Child #2 | 1024 | 256 | 256 | 1.25 | 4000 | 800 |

**Table 4.** Input parameters of the precursor runs.

| | $\overline{U}_0$, ms$^{-1}$ | $dp/dx$, Pa m$^{-1}$ | $z_0$, m | $T_s$, K | $\overline{w'\theta'}$, K ms$^{-1}$ | $dT_s/dt$, Ks$^{-1}$ | Run time, s |
|--|------|------|------|------|------|------|------|
| NBL (coarse) | 13.8 | $-2 \times 10^{-4}$ | $1.2 \times 10^{-3}$ | 300 | 0 | — | 144 000 |
| NBL (fine) | 14.0 | $-2 \times 10^{-4}$ | $1.6 \times 10^{-3}$ | 300 | 0 | — | 172 800 |
| CBL | 11.5 | $-1 \times 10^{-4}$ | $5 \times 10^{-4}$ | 281 | 0.015 | — | 525 600 |
| SBL | 13.0 | $-5 \times 10^{-4}$ | $8 \times 10^{-4}$ | 300 | — | -0.2 | 259 200 |

precursor domain is simulated with cyclic boundaries until the flow reaches a steady state. The resulting mean wind speed and temperature profiles are then copied over the larger main domain to set up an initial non-cyclic flow with a developed turbulence. Provided that the main run is simulated with the same forcing as the precursor, the mean profiles in the main run remain stationary.

The size of the precursor domain is usually smaller than for the main run, and the y-shift procedure is performed at left/right cyclic boundaries to avoid non-physical regularity of the flow (Munters et al., 2016). The y-shift procedure is also applied in the main run for an additional disruption of regularity. Using the precursor-main run scheme also ensures that an idealized input flow remains the same within a stability case regarded.

The grid characteristics of the root and innermost child domain in the PALM simulation were selected to closely match the SOWFA simulation in Nybø et al. (2020). The ratio between the parent and child domains' grid spacing, thus, would reach 8 (from 10 m to 1.25 m for NBL and CBL cases) or 4 (from 5 m to 1.25 m for SBL case). As shown by Hellsten et al. (2021), the discrepancy with a fine-grid simulation in PALM increases if the grid spacing ratio is 4 or higher. Therefore, we add intermediate child domains and reduce the grid spacing by a factor of 2 until the desired refinement is reached. Hence, NBL and CBL simulations contain three child domains, while the SBL simulation has two (Table 2, 3, Fig. 1).

We perform one-way and two-way nested simulations. To evaluate the nesting effect, we also simulate domains without nested grids using the same precursor flow. Due to high computational time and memory requirements, we only simulate non-nested domains for the grid spacing of $\Delta_x = 10$ m and 5 m.

**Table 5.** Steady state of the precursor runs – turbulent inflow for the main run.

|  | $\overline{U}_{119}$, ms$^{-1}$ | TI$_8$0, % | $T_s$, K | $L$, m | Capping inversion, K/100 m |
|---|---|---|---|---|---|
| NBL (coarse) | 12.3 | 7.5 | 300 | $10^6$ | 0 |
| NBL (fine) | 12.6 | 7.7 | 300 | $10^6$ | 0 |
| CBL | 12.1 | 6.2 | 295 | -333 | 7.4 |
| SBL | 12.8 | 4.6 | 291 | 529 | 9 |

The precursor profiles undergo development during a simulation and thus may deviate from the initial profiles. The precursor's input parameters are then selected so that the resulting steady-state profiles of mean wind speed and turbulence intensity follow the values estimated from the measurements, particularly the wind speed at the reference height. The Coriolis force is switched off; hence the required wind speed and turbulence intensity profiles in the precursor run are enforced by a combination of the parameters: the initial mean wind $\overline{U}_0$, the pressure gradient forcing $dp/dx$, and the roughness length $z_0$. The NBL case is run as the true neutral flow with no heat flux. The CBL case is defined via the positive heat flux $\overline{w'\theta'}$ in addition to the parameters mentioned above. The SBL case uses surface cooling over time $dT_s/dt$ instead of the heat flux (Wurps et al., 2020). NBL and SBL cases start with zero temperature gradient; CBL case has an initial temperature gradient of $1\,\text{K}/100\text{m}$. The surface temperature $T_s$ is varied to match the conditions observed during the reference meteorological measurements at FINO1. The precursor domain characteristics and input parameters are listed in Tables $2-4$.

During the precursor simulation, the initial profiles are altered due to the influence of pressure forcing and heat fluxes. The resulting precursor profiles are provided in Table 5; the same profiles are used to initialize the main run.

We run main simulations for three hours with a dynamic time step selected by the model. The simulation is then continued for another hour with the fixed time step of $\Delta t = 0.05\,\text{s}$ to obtain a high-frequency output. Then, we probe time series of each wind speed component at the center of the innermost child domain and the corresponding points of the parent domain (Fig. 1). The high-frequency time series are further used to compare turbulence statistics with the measurements. Spatial averages (cross-sectional flows, profiles) are calculated for 10-minute periods.

### 3.3 Turbulence characteristics

We evaluate the model performance based on turbulence characteristics: power spectrum, coherence, co-coherence, and phase. The coherence represents a correlation between time series $a(t)$ and $b(t)$ at two points separated by a certain distance $\delta$ and is calculated as follows

$$\text{Coh}_{ab} = \frac{S_{ab}}{\sqrt{S_{aa}S_{bb}}} \tag{4}$$

where $S_{aa}$ and $S_{bb}$ are the spectral densities of $a(t)$ and $b(t)$, while $S_{ab}$ is the cross-spectrum of the same series.

The co-coherence represents the real part of the coherence

$$\text{Co}_{ab} = \text{Re}\,\text{Coh}_{ab} = \text{Re}\,\frac{S_{ab}}{\sqrt{S_{aa}S_{bb}}} \tag{5}$$

The phase $\phi_{ab}$ shows the level of synchronicity between time series $a(t)$ and $b(t)$

$$\phi_{ab} = \arctan \frac{\text{Re Coh}_{ab}}{\text{Im Coh}_{ab}} \tag{6}$$

Since the measurement time series are available only for three levels: 40, 60, and 80 m, the spectra are calculated and compared at $h = 80$ m for the total horizontal $U = \sqrt{u^2 + v^2}$ and vertical $w$ wind speed. The co-coherence is calculated for two vertical separations of $\delta = 20$ m (between levels 60 and 80 m) and $\delta = 40$ m (between levels 40 and 80 m). The sampling frequency for the LES time series matches the output frequency $f_s^{LES} = 1/0.05\,\text{s} = 20\,\text{Hz}$, and the segment length is chosen as 60 s. The sampling frequency for the measurement time series is lower $f_s^{mast} = 1/0.1\,\text{s} = 10\,\text{Hz}$, although the segment length is left the same.

### 3.4 Flow characteristics for load analysis

We also review flow characteristics relevant to the turbine performance analysis: power law coefficient and turbulence anisotropy.

The power law is commonly applied to assess wind resources at the hub height from near-surface wind speed measurements.

$$U(z) = \overline{U}_{10} \left( \frac{z}{10} \right)^{\alpha} \tag{7}$$

where $\overline{U}_{10}$ is the wind speed at $z = 10$ m and $\alpha$ is the power law exponent. The power law exponent is sensitive to atmospheric conditions and is usually approximated with a constant, e.g., $\alpha = 1/7$ is applicable to neutral onshore sites but not other stabilities (Touma, 1977). Often, the approximations do not reflect seasonal and diurnal variations in mean wind profiles (Bratton and Womeldorf, 2011; Jung and Schindler, 2021). Hence, simulating a long time series with the LES gives a possibility to study wind profiles in detail.

The anisotropic turbulence naturally develops in a simulation with an anisotropic grid resolution (Haering et al., 2019), but may also occur in isotropic grids, such as those used in this study. The anisotropic turbulence affects wind turbine loads, particularly fatigue loads. Therefore, it is important to evaluate its strength in the simulation (Dimitrov et al., 2017). We estimate turbulence anisotropy by comparing spectra of velocity components for the normalized frequency $f_n = fz/U_z$, where $z = 80$ m and $U_z$ is the horizontal velocity at this level. We compute ratios $S_{vv}/S_{uu}$ and $S_{ww}/S_{uu}$ for all regarded cases at $f_n \approx 1$. The closer both ratios are to the theoretical value of $4/3 = 1.333$, the more isotropic is the simulated turbulence (Weiler and Burling, 1967; Smedman et al., 2003).

## 4 Results

### 4.1 Nesting effects

All LESs are run at 1024 cores for each case with a time step of $\Delta t = 0.05$ s; the required simulation times for each scenario are summarized in Table 6. Since the domains vary in size and number of grid points, we compare not the total CPU time but

**Table 6.** CPU time in seconds used per second of simulated time. All simulations run at 1024 cores with a time step of $\Delta t = 0.05\,\mathrm{s}$

| Stability | $\Delta_x$, m | non-nested | one-way | two-way |
|-----------|------|------------|---------|---------|
| NBL | 10 | 5.1 | 18.4 | 20.9 |
| NBL | 5 | 31.7 | - | - |
| CBL | 10 | 7.9 | 28.8 | 30.8 |
| SBL | 5 | 4.5 | 25.1 | 28.7 |

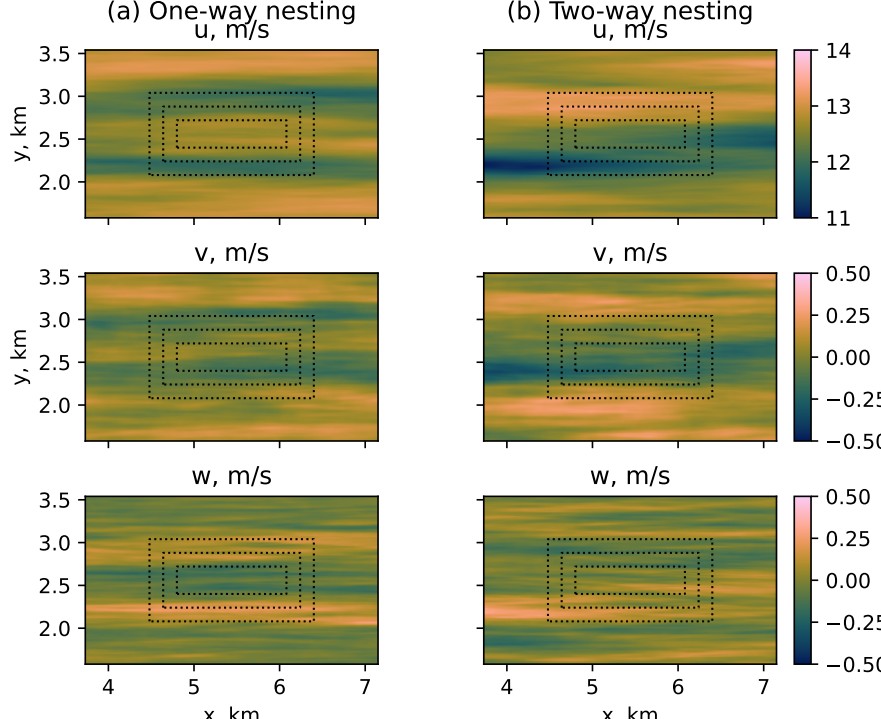

**Figure 2.** NBL, flow at the reference height of 119 m for different wind speed components, (a) one-way nesting, (b) two-way nesting.

CPU time per second of the simulated time. The non-nested coarse domain ($\Delta_x = 10\,\mathrm{m}$) is not computationally demanding, regardless of the stability case. However, the required CPU time gradually increases if the grid spacing is reduced globally for the whole domain. As could be seen for the NBL case, the CPU time per second of the simulated time increases from 5.1 s for $\Delta_x = 10\,\mathrm{m}$ to 31.7 s for $\Delta_x = 5\,\mathrm{m}$, respectively. Refining the grid locally by adding child domains increases the CPU time compared to the coarse reference non-nested grid ($\Delta_x = 10\,\mathrm{m}$). Still, the nested simulation finishes faster than the globally refined non-nested simulation ($\Delta_x = 5\,\mathrm{m}$), while allowing better a local grid refinement up to $\Delta_x = 1.25\,\mathrm{m}$.

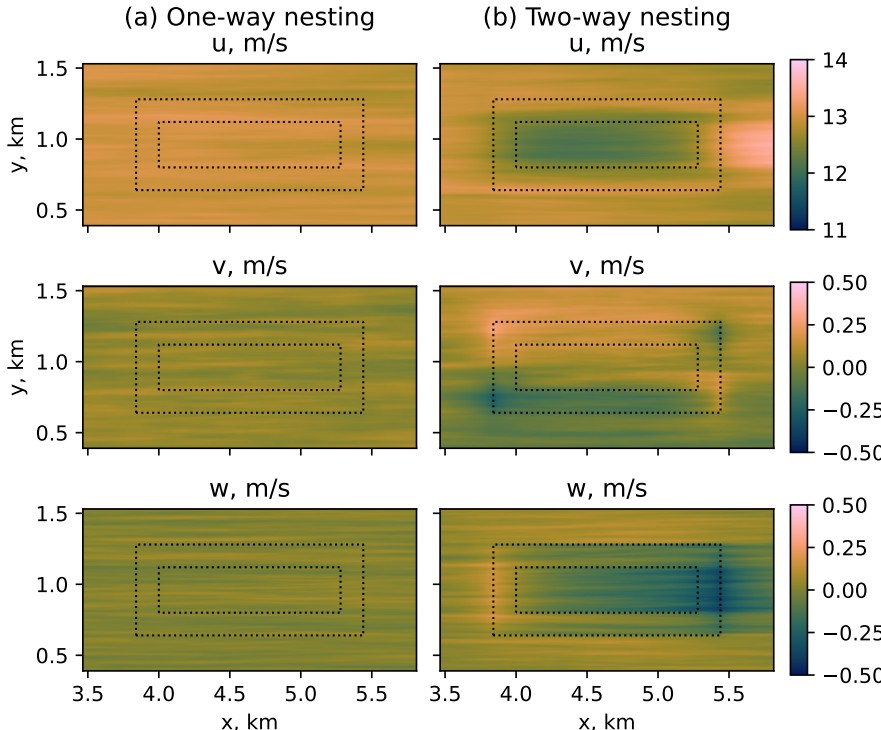

**Figure 3.** SBL, flow at the reference height of 119 m for different wind speed components, (a) one-way nesting, (b) two-way nesting.

Both NBL and CBL simulations have the same domain structure and grid spacing (Table 2). However, CBL simulations require more CPU time compared to the respective NBL (true neutral) simulations due to solving the temperature equation. SBL simulations use CPU time comparable to NBL simulations due to having one child domain less and a smaller root domain size – and thus a lower overall number of the grid points (Table 3).

Two-way nested simulations require additional $\sim 2 - 3\,\mathrm{s}$ of the CPU time per simulated time step to anterpolate the child domain solution back to the parent domain. This results in about 10% increase of the CPU time compared to one-way nesting. It should be noted that, unless obtaining high-frequency time series is the main goal of a simulation, the time step can be gradually increased for non-nested runs in order to speed up the computation. The computational time will, nevertheless, increase in a similar proportion with the global grid refinement. The time step in nested runs is still limited by the lowest grid spacing in child domains. E.g., the dynamic step in the regarded configuration does not exceed $0.075\,\mathrm{s}$ to satisfy Courant–Friedrichs–Lewy condition.

Depending on the simulation conditions, LES produces different results in the nested area. If the true neutral case is defined in PALM explicitly via setting a corresponding flag, the one-way and the two-way nested simulations behave similarly with respect to grid spacing and feedback between domains (Fig. 2). Switching on the true neutral flag means that the temperature

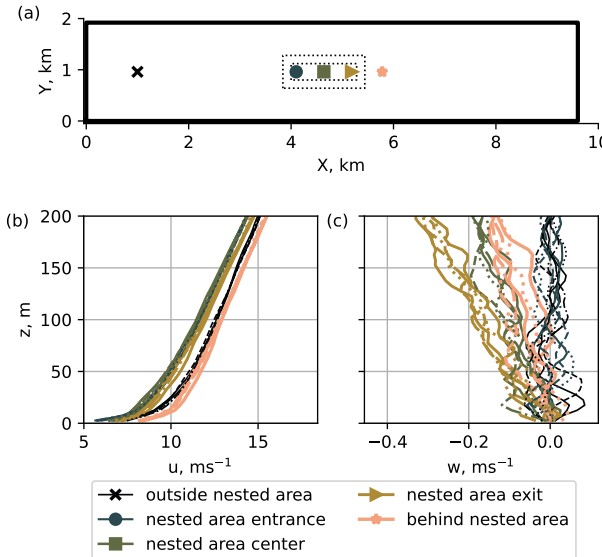

**Figure 4.** 10-minute average profiles, SBL two-way nested case. (a) Sampling points; (b) the mean flow is slowed down in the nested area; (c) the vertical flow near the entrance of the nested area remains weak but becomes stronger as the flow passes through the nested area.

equation and buoyancy terms are not considered in the calculations. As long as those terms are introduced for non-neutral simulations, the two-way nested simulation results in a decreased flow speed in child domains.

Since the child domains anterpolate their solution back to the parent domain, the area of reduced flow speed spreads to the root domain. While the effect is less prominent for the instantaneous fields, it becomes apparent in the 10-minute averaged flow (Fig. 3). The induction of downward vertical wind in two-way nested simulations was already described by Hellsten et al. (2021) for the 5-hour averaged buoyancy-driven flow in PALM. Hellsten et al. (2021) argued that the effect of the secondary circulation described by Moeng et al. (2007) was caused solely by the insufficient domain size and explained it with the

different grid spacing and subsequent divergence of the vertical heat flux in the parent and child domains. The researchers hypothesized that the secondary circulation was an inevitable side effect of the two-way nesting solution due to the better resolution of the turbulence mixing in child domains. In the case of the shear-driven flow, we observe that the slowing effect is more prominent and develops faster. The effect emerges in the beginning of the simulation within 20 minutes – an approximate time required for the precursor flow to pass the main run domain. In addition, some of the quantities of a shear-driven flow,

mainly the vertical velocity $w$, are not uniformly distributed inside the child domains (Fig. 4).

## 4.2 Subgrid scales

LES resolves scales larger than the grid spacing directly but approximates smaller scales. In a well-resolved flow, the unresolved (subgrid) scales should not exceed the resolved ones. This relation holds for all simulations performed, implying that the grid spacing of $\Delta = 10\,\mathrm{m}$ is already small enough for the given flow (Fig. 5). The grid refinement does not strongly affect

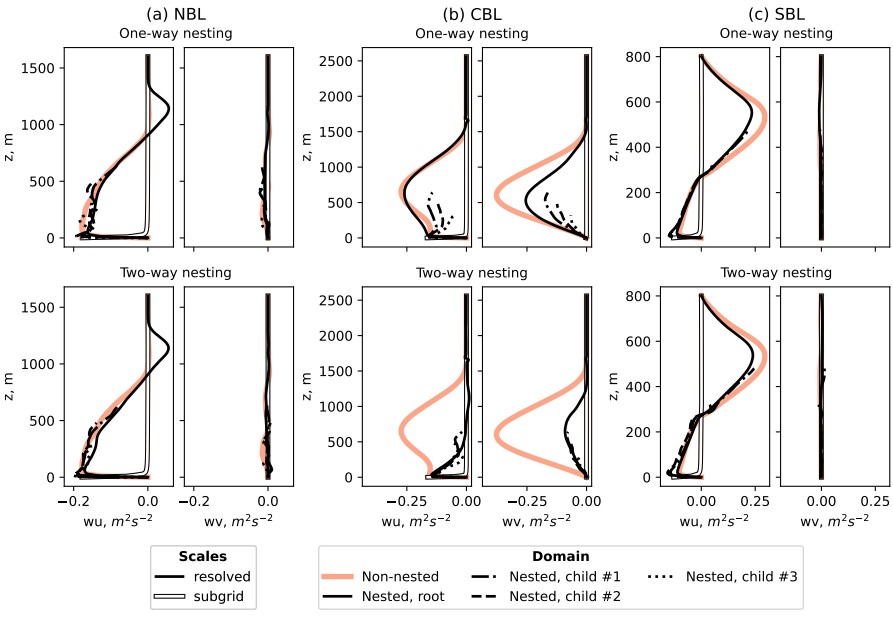

**Figure 5.** Comparison of resolved and subgrid-scale momentum fluxes for different stability simulations and nesting modes

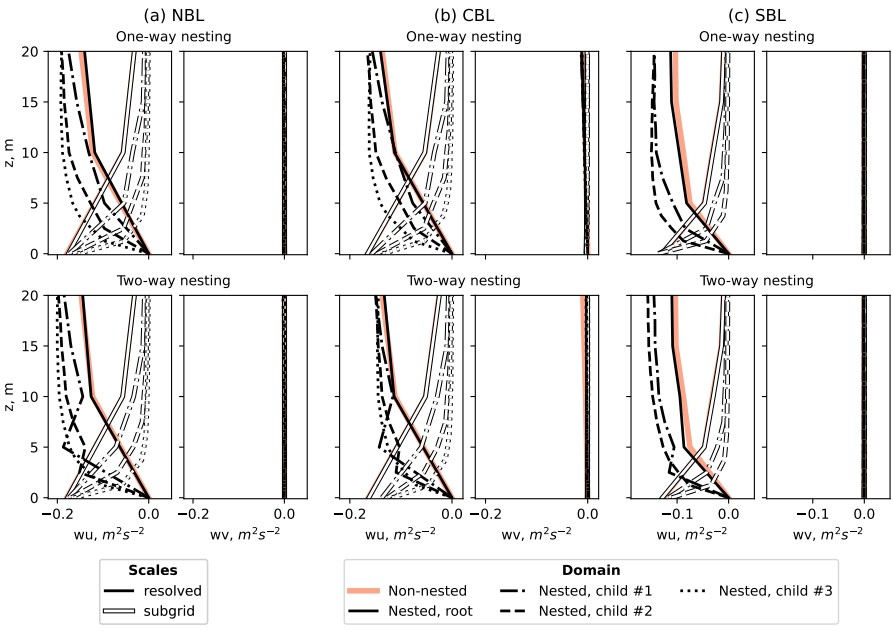

**Figure 6.** Comparison of near-surface resolved and subgrid-scale momentum fluxes for different stability simulations and nesting modes

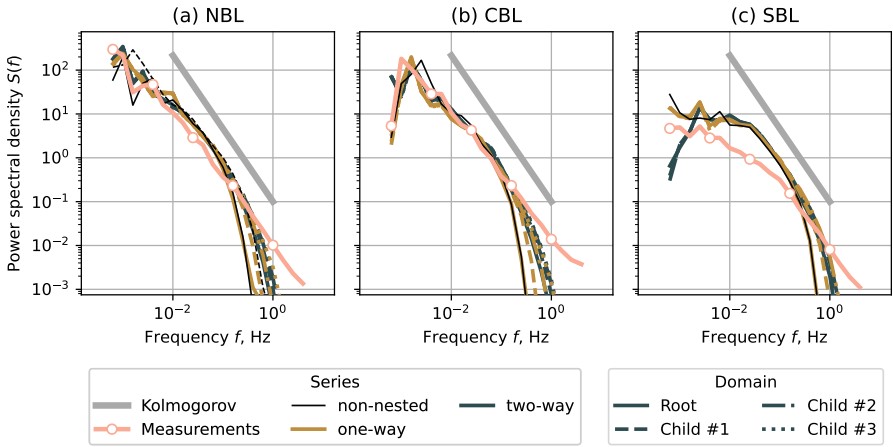

**Figure 7.** Spectra for the horizontal velocity $u$ at the height $z = 80\,\mathrm{m}$. (a) NBL case, (b) CBL case, (c) SBL case.

momentum fluxes, except for the CBL case (Fig. 5b), where turbulent eddies are generally larger than in the NBL and SBL cases. The effect from the nesting mode is also the most pronounced in CBL simulations (Fig. 5b). The resolved $\overline{wu}$ and $\overline{wv}$ fluxes remain stationary in the one-way nesting mode, but decrease over time in the two-way nesting mode and eventually merge.

The subgrid-scale fluxes consistently remain near zero for all levels except near-surface cells, where the turbulence intensity is expected to be high due to the surface influence 6. Consequently, the near-surface subgrid-scale fluxes are comparable to resolved-scale fluxes. However, the subgrid-scale fluxes at lower levels tend to zero faster as the grid spacing is refined. Unlike the one-way nesting mode, the resolved fluxes in the two-way nesting mode show a non-monotonic behavior near the surface in the intermediate child domains. The effect is observed in all two-way simulations, including true neutral conditions. Therefore, it cannot be solely caused by the flow difference in the nested and non-nested areas, despite the flux profiles being time and spatial averages. The occurring non-monotonic behavior can be rather attributed to the way PALM performs anterpolation from a child to the parent domain.

### 4.3 Turbulence characteristics

Since the flow is driven by the pressure gradient instead of the Coriolis force, the flow is aligned with the $x$-axis, and the wind direction remains nearly constant. The fluctuations of the lateral component $v$ are stronger for the measurement time series. Therefore, we compare turbulence statistics of the horizontal wind speed $u$ from the LES results to the total horizontal flow in the measurements $U = \sqrt{u^2 + v^2}$ and omit the lateral component $v$ for the LES data.

In one-way nested simulations, the parent domain does not receive feedback from the child domain. Consequently, the spectral characteristics of non-nested domains with the grid spacing of $\Delta_x = 10\,\mathrm{m}$ (NBL and CBL) and $5\,\mathrm{m}$ (SBL) match the characteristics of the corresponding domain in a one-way nesting simulation (Fig. 7, 8). The individual spectra of the nested

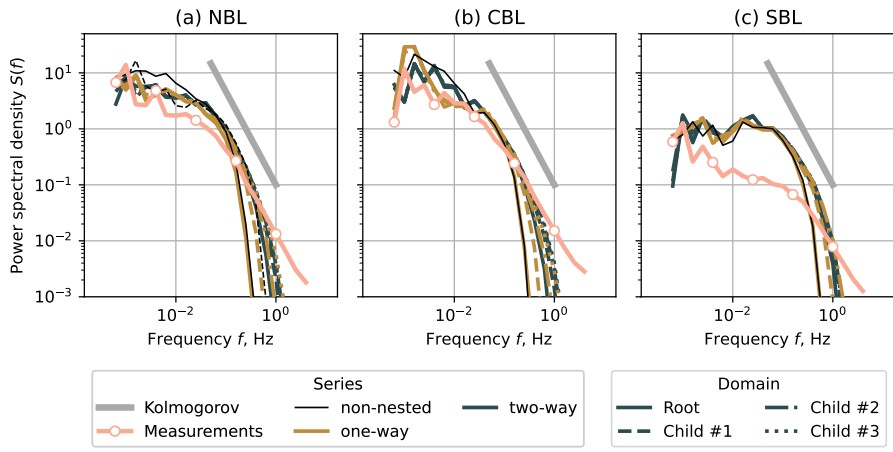

**Figure 8.** Spectra for the vertical velocity $w$ at the height $z = 80\,\mathrm{m}$. (a) NBL case, (b) CBL case, (c) SBL case.

domains lay apart from each other but show improvement as the grid spacing is reduced. The inertial subrange resolved by LES widens as the grid becomes more refined; however, it is not fully resolved despite the grid spacing being reduced to $\Delta_x = 1.25\,\mathrm{m}$.

The two-way nesting mode ensures feedback between the nested domains. Therefore, the root and child domain spectra lie closer to each other and to the one-way spectra of the most refined child domain ($\Delta_x = 1.25\,\mathrm{m}$). Despite the exchange between domains in the two-way nested case, the spectral characteristics do not coincide perfectly. The inertial subrange being shorter for $\Delta_x = 10\,\mathrm{m}$ than for the refined domains implies that the grid resolution is the limiting factor, and the solution for the root domain cannot be improved further even in the two-way nesting case.

Despite the NBL case being simulated as a true neutral condition, it showed good agreement with the measurements on par with the CBL case. The result suggests that it is possible to omit a weak heat flux in neutral cases to save computational time and avoid secondary circulation in the two-way nesting mode.

The SBL simulations largely overestimate the energy contained in low-frequency eddies. The inertial subrange of the corresponding measurement time series also starts at higher frequencies, unlike in the NBL and CBL cases. The LES does not fully resolve high frequencies despite gradually reduced grid spacing. Hence the overall agreement for the SBL case is worse than for NBL and CBL. When comparing available measurement profiles for the specific period of SBL time series, we did not observe anomalies or irregularities, such as reported by Kettle (2014), which could be studied as a possible cause of a discrepancy. The existing studies on SBL simulations with PALM (Beare et al., 2006; Wurps et al., 2020) do not compare simulated spectra against measurements, but evaluate other aspects, such as fluxes and grid resolution influence. Hence, simulating SBL in PALM may require additional studies focusing on turbulence characteristics.

In order to match the SBL spectra shape, we performed a short SBL simulation with lower forcing, which lead to a decreased turbulence intensity but stronger mean profile shear. The results are provided in Appendix.

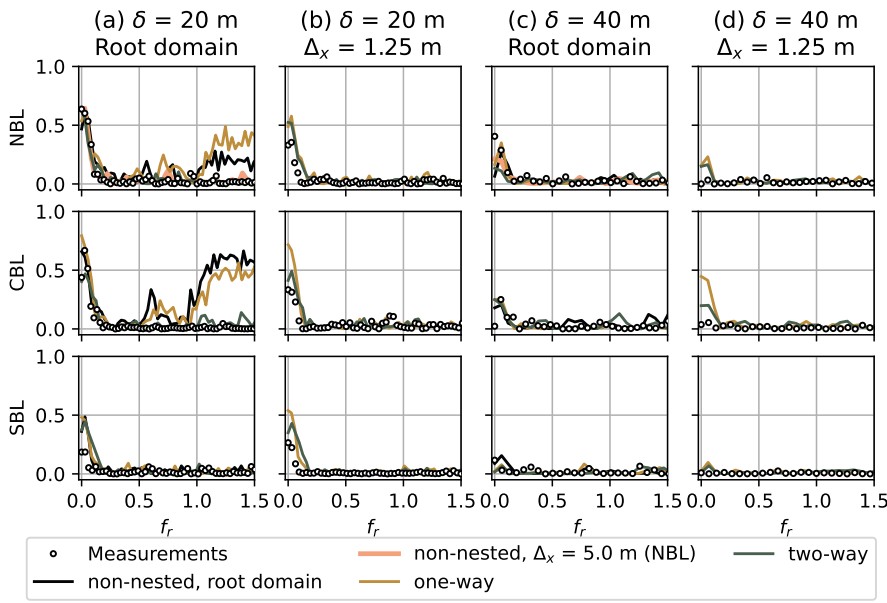

**Figure 9.** Coherence for the horizontal velocity $u$ and different stability cases. (a) Root domain ($\Delta_x = 10\,\mathrm{m}$ for NBL and CBL, $\Delta_x = 5\,\mathrm{m}$ for SBL), vertical separation $\delta = 20\,\mathrm{m}$. (b) Innermost child domain ($\Delta_x = 1.25\,\mathrm{m}$, all cases), vertical separation $\delta = 20\,\mathrm{m}$. (c) Root domain ($\Delta_x = 10\,\mathrm{m}$ for NBL and CBL, $\Delta_x = 5\,\mathrm{m}$ for SBL), vertical separation $\delta = 40\,\mathrm{m}$. (d) Innermost child domain ($\Delta_x = 1.25\,\mathrm{m}$, all cases), vertical separation $\delta = 40\,\mathrm{m}$.

The coherence, co-coherence, and phase are plotted against the reduced frequency

$$f_r = \frac{f\delta}{\overline{u}} \tag{8}$$

where $f$ is the original frequency, $\delta$ is the vertical separation distance and $\overline{u}$ is the mean wind speed of the two regarded levels: $60\,\mathrm{m}$ and $80\,\mathrm{m}$ for $\delta = 20\,\mathrm{m}$, or $40\,\mathrm{m}$ and $80\,\mathrm{m}$ for $\delta = 40\,\mathrm{m}$.

The coherence and co-coherence calculated for NBL and CBL coarse domains ($\Delta_x = 10\,\mathrm{m}$) and $\delta = 20\,\mathrm{m}$ show strong deviation from the measurements for the one-way and non-nested simulations at $f_r > 1$ (Fig. 9a, Fig. 10a). The tendency to the coherence/co-coherence value of $0.5$ suggests that the time series at points separated by $\delta = 20\,\mathrm{m}$ remain partially correlated in the coarse grid, which is not the case for the corresponding measurements. While the most refined child domain ($\Delta_x = 1.25\,\mathrm{m}$) shows a good match between the LES and measurement series (Fig. 9b, 10b), the agreement already improves

for $\Delta_x = 5\,\mathrm{m}$, and the correlation falls to zero for $f_r > 0.5$. The SBL case shows better agreement for the root domain because of the lower initial grid spacing $\Delta_x = 5\,\mathrm{m}$. Nevertheless, the coherence is noticeably overestimated for low $f_r$ compared to the measurements (Fig. 9ab). The time series are generally uncorrelated for the vertical separation of $\delta = 40\,\mathrm{m}$ both for the LESs and measurements (Fig. 9cd, Fig. 10cd). However, the NBL case does not capture the high coherence value at $f_r = 0$ observed in the measurements.

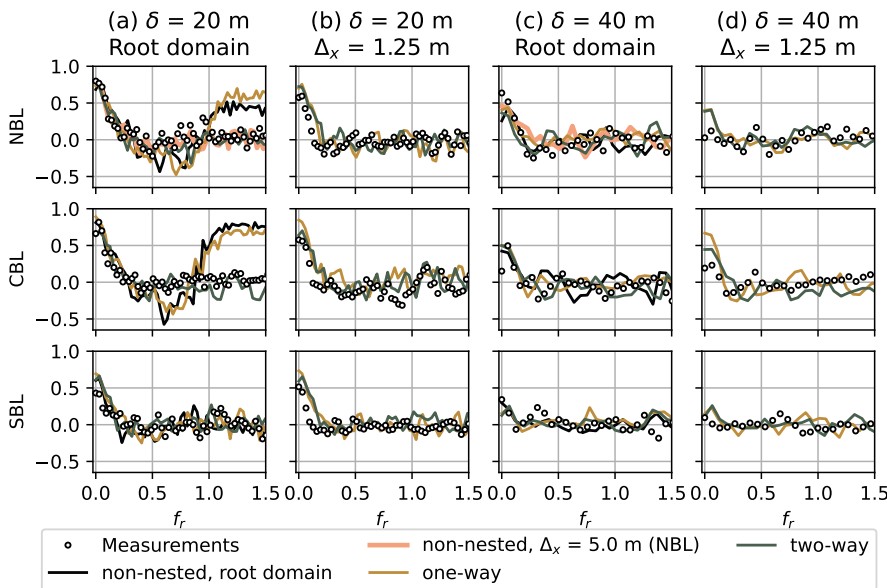

**Figure 10.** Co-coherence for the horizontal velocity $u$ and different stability cases. (a) Root domain ($\Delta_x = 10\,\text{m}$ for NBL and CBL, $\Delta_x = 5\,\text{m}$ for SBL), vertical separation $\delta = 20\,\text{m}$. (b) Innermost child domain ($\Delta_x = 1.25\,\text{m}$, all cases), vertical separation $\delta = 20\,\text{m}$. (c) Root domain ($\Delta_x = 10\,\text{m}$ for NBL and CBL, $\Delta_x = 5\,\text{m}$ for SBL), vertical separation $\delta = 40\,\text{m}$. (d) Innermost child domain ($\Delta_x = 1.25\,\text{m}$, all cases), vertical separation $\delta = 40\,\text{m}$.

The phase plots are in line with the coherence. The time series are in-phase for $f_r < 0.1$, where the coherence is above zero. The effect is strong for the low vertical separation of $\delta = 20\,\text{m}$ (Fig. 11ab) and is in good agreement with the measurements. The phase becomes more chaotic as the vertical separation distance increases to $\delta = 40\,\text{m}$ (Fig. 11cd), while the time series become less correlated (Fig. 9cd, 10cd).

## 4.4 Other flow characteristics

### 4.4.1 Power law

In general, the power law coefficient follows the known trend, also observed in the measurement profile fits (Table 7): high value in the stable layer and low value in the convective layer (Touma, 1977). The discrepancy between exact values of $\alpha$ in measurement and simulated fits is primarily caused by the different way of obtaining $U_{10}$. For sonics data, $U_{10}$ is calculated from the previously estimated profile Eq. (1). The LES returns full mean profile on the pre-defined grid, so $U_{10}$ can be interpolated to the level of $z = 10\,\text{m}$. $U_{10}$ derived from LES data consistently deviates from measurements $U_{10}$ by $10-20\%$, thus affecting the estimation of the power law exponent.

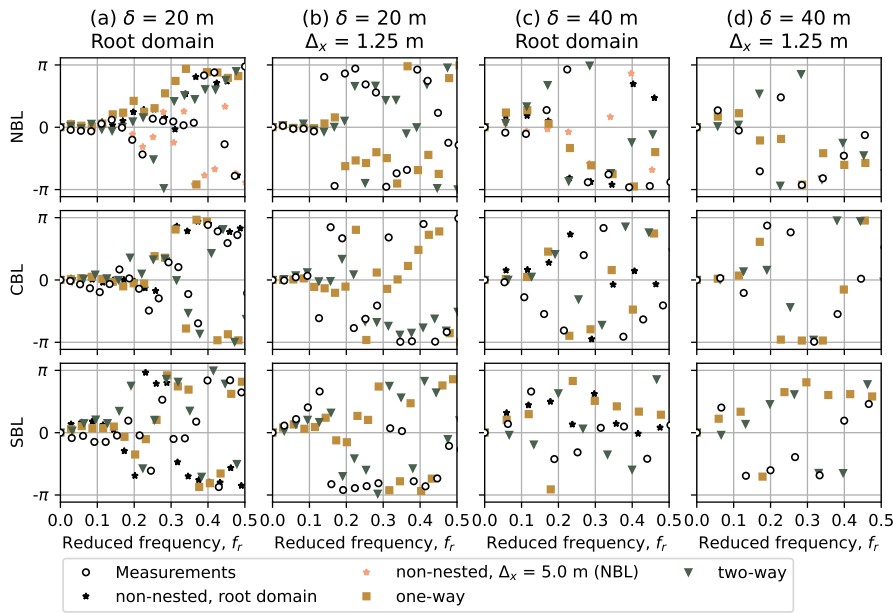

**Figure 11.** Phase plot for the horizontal velocity $u$ and different stability cases and domains. (a) Root domain ($\Delta_x = 10\,\text{m}$ for NBL and CBL, $\Delta_x = 5\,\text{m}$ for SBL), vertical separation $\delta = 20\,\text{m}$. (b) Innermost child domain ($\Delta_x = 1.25\,\text{m}$, all cases), vertical separation $\delta = 20\,\text{m}$. (c) Root domain ($\Delta_x = 10\,\text{m}$ for NBL and CBL, $\Delta_x = 5\,\text{m}$ for SBL), vertical separation $\delta = 40\,\text{m}$. (d) Innermost child domain ($\Delta_x = 1.25\,\text{m}$, all cases), vertical separation $\delta = 40\,\text{m}$.

**Table 7.** Estimated power law coefficient.

| | | Power law coefficient $\alpha$ | | |
|---|---|---|---|---|
| Nesting | $\Delta_x$, m | NBL | CBL | SBL |
| non-nested | 10 | 0.111 | 0.093 | - |
| non-nested | 5 | 0.099 | - | 0.154 |
| one-way | 10 | 0.112 | 0.093 | - |
| one-way | 5 | 0.103 | 0.067 | 0.156 |
| one-way | 2.5 | 0.092 | 0.077 | 0.145 |
| one-way | 1.25 | 0.087 | 0.073 | 0.145 |
| two-way | 10 | 0.109 | 0.089 | - |
| two-way | 5 | 0.095 | 0.083 | 0.158 |
| two-way | 2.5 | 0.088 | 0.080 | 0.164 |
| two-way | 1.25 | 0.085 | 0.077 | 0.172 |
| Measurements | | 0.061 | 0.023 | 0.237 |

| | | NBL | | CBL | | SBL | |
|---|---|---|---|---|---|---|---|
| | | $S_{vv}/S_{uu}$ | $S_{ww}/S_{uu}$ | $S_{vv}/S_{uu}$ | $S_{ww}/S_{uu}$ | $S_{vv}/S_{uu}$ | $S_{ww}/S_{uu}$ |
| non-nested | 10 m | 1.382 | 1.591 | 1.436 | 1.532 | | |
| non-nested | 5 m | 1.357 | 1.001 | | | 1.567 | 1.092 |
| one-way | 10 m | 1.841 | 1.367 | 1.426 | 1.507 | | |
| one-way | 5 m | 1.580 | 1.198 | 1.514 | 1.666 | 1.545 | 0.916 |
| one-way | 2.5 m | 1.560 | 1.235 | 1.219 | 1.547 | 1.304 | 0.804 |
| one-way | 1.25 m | 1.482 | 1.191 | 1.209 | 1.470 | 1.431 | 0.808 |
| two-way | 10 m | 1.361 | 1.206 | 1.252 | 1.547 | | |
| two-way | 5 m | 1.151 | 1.000 | 1.595 | 1.401 | 1.043 | 0.915 |
| two-way | 2.5 m | 1.164 | 1.004 | 1.693 | 1.433 | 0.641 | 0.294 |
| two-way | 1.25 m | 1.217 | 1.045 | 1.704 | 1.438 | 1.032 | 1.012 |
| measurements | | 0.948 | 1.045 | 1.604 | 0.969 | 0.866 | 0.359 |

**Figure 12.** Comparison of anisotropy across the regarded stability and nesting cases. The colormap is centered at the value $4/3 = 1.333$.

The estimated power law coefficient $\alpha$ shows little variation for the NBL and CBL domains of the same refinement, but implies high sensitivity of the SBL profiles. Considering higher shear in the SBL profiles, the grid refinement may affect the estimation of $U_{10}$ stronger than lower shear NBL and CBL profiles.

### 4.4.2 Turbulence anisotropy

The anisotropy estimation captures only general trends seen in the measurements with the nesting modes being radically different between each other (Fig. 12). Since the inertial subrange resolved in a one-way nested root domain is slightly shorter than of a two-way root domain (Fig. 7−8), $f_n \approx 1$ may fall outside of the resolved subrange and provide a less precise estimation. The two-way nested cases approach closer to the anisotropy seen in the measurement, although the anisotropy strength may not match the value seen in the measurement data. The divergence is particularly strong for the SBL simulation, primarily caused by the differences in power density spectra discussed in Sec. 4.3.

## 5 Conclusions

We performed nested LES of three stability cases for the horizontal mean wind speed of $12-13\,\mathrm{m\,s^{-1}}$ at the reference height of 119 m. The simulations were verified by comparing turbulence characteristics to the corresponding measurement time series. The comparison showed that the grid spacing of $\Delta_x = 10\,\mathrm{m}$ was insufficient for NBL and CBL simulations; the spectral and coherence characteristics had improved their agreement with the measurements after the spacing was reduced to $\Delta_x = 5\,\mathrm{m}$ via

nesting or a refined non-nested domain simulation. The inertial subrange was not fully resolved despite further refinement and remained narrower than for the measurement time series even at $\Delta_x = 1.25\,\mathrm{m}$.

We confirmed that the nesting mode does not affect the true neutral simulation, unlike when the temperature equation is solved along with other prognostic equations for CBL and SBL conditions. In the case of CBL or SBL, the flow inside the child domain differed for the one-way and two-way nesting. The two-way nested simulation produced a secondary circulation resulting in a decreased velocity and increased turbulence intensity in the child domains. Due to a strong horizontal shear, the irregularities in lateral and vertical velocity profiles were spread non-uniformly, e.g., the downward flow was stronger at the exit of the nested domain. The horizontal flow accelerated after leaving the nested area so that the mass conservation law was not violated eventually. Unlike the existing research on buoyancy-driven flows, the two-way nesting effects in a shear-driven flow emerged in the first hour of the LES and did not dissipate as the simulation proceed for three more hours.

In theory, the two-way nesting is a good option to refine the grid in the area of interest of a non-homogeneous flow, e.g., wind turbine wakes, as the feedback between parent and child domain allows accounting the irregularities after the flow exits the nested area. However, the fast development of a secondary circulation in the shear-driven flow limits the two-way nesting application strictly to the true neutral condition. The one-way nested simulation did not add anomalies to the flow; each child domain only refined the grid spacing and resolved small turbulence scales. We, therefore, recommend using the one-way nesting mode for the wind turbine wake simulation. In the case when the two-way nesting mode is preferable, only a true neutral setup does not produce secondary circulation.

*Code and data availability.* The PALM model system is freely available at https://palm.muk.uni-hannover.de (last access: October 12, 2022) and distributed under the GNU General Public License v3 (http://www.gnu.org/copyleft/gpl.html, last access: October 12, 2022). The LESs in this article were performed using PALM model system v21.10. The corresponding version is provided at https://doi.org/10.5281/zenodo.7886678 (Krutova, 2022) together with input and output files, post-processing scripts needed to reproduce the figures. The processed high-frequency sonic anemometer are available upon request after the permission from DEWI (Deutsches Windenergie Institut) is granted.

## Appendix A: SBL simulation with reduced forcing

We performed a test simulation of an SBL precursor for the same wind speed but weaker pressure gradient (-0.0001 Pa/m instead of -0.0005 Pa/m) and slightly stronger surface cooling (-0.3 K/s instead of -0.2 K/s). As a result of the decreased forcing, the developed profiles deviated from the reference measurements and showed stronger shear but lower turbulence intensity (Fig. A1). Due to the computational time constraints we simulate only a non-nested main run for a comparison of spectral characteristics. We observe a better agreement with the measurements spectra (Fig. A2), especially in the $w$-component, which spectrum does not follow $-5/3$ theoretical slope. Therefore, we are able to match only one of two: either SBL profiles or SBL spectra – and observe a strong discrepancy in another.

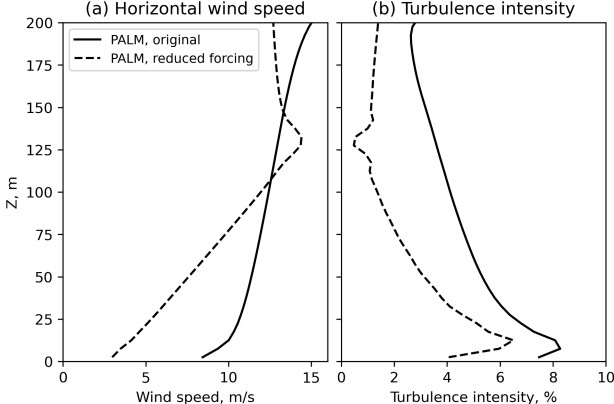

**Figure A1.** Precursor run profiles with original and reduced pressure forcing. (a) Horizontal flow mean profile, (b) turbulence intensity profile

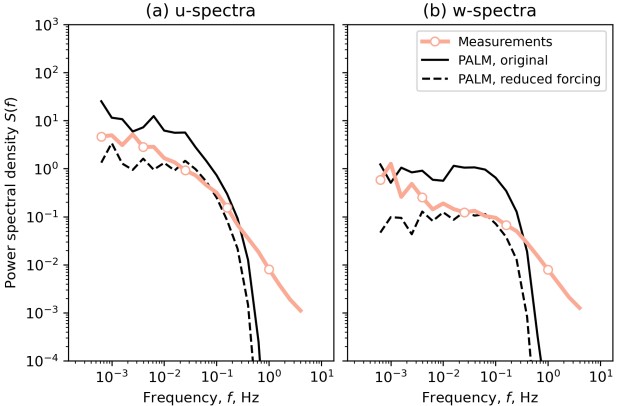

**Figure A2.** Main run spectra with original and reduced pressure forcing. (a) Horizontal velocity spectrum, (b) vertical velocity spectrum

*Author contributions.* MK performed the LES simulations and analysis in accordance to the plan developed by MPB; JR and FGN provided valuable discussion on explaining the discrepancies with the measurement data.

*Competing interests.* The authors declare that they have no conflict of interest.

*Acknowledgements.* The authors would like to thank DEWI (Deutsches Windenergi Institut) for providing the FINO1 high-resolution sonic anemometer data and Astrid Nybø from University of Bergen for the additional information and guidance.

The large-eddy simulations for this study have been performed by using the high performance computer facilities of the Norwegian e-infrastructure Uninett Sigma2 (project number NS9696K).

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
