# Peer review of "Self-nested large-eddy simulations in PALM Model System v21.10 for offshore wind prediction under different atmospheric stability conditions"

_Geoscientific Model Development, 2022_

## Referee Comment (RC1)

**Referee comment on "Self-nested large-eddy simulations in PALM Model System v21.10 for offshore wind prediction under different atmospheric stability conditions"**

This article presents a comparison of self-nested large-eddy simulations using PALM with the field measurements from FINO1 offshore met mast. The topic of localized mesh refinement to reduce the computational expense, while resolving smaller turbulent scales is of interest to the scientific community. Therefore, evaluating existing modeling frameworks for such applications is important and falls within the scope of the journal.

Overall, the authors have done a decent job in evaluating the model and the results are presented in a clear manner. I do have some comments, which I believe should be addressed before the article is considered for publication:

**Major Comments:**

1. The primary motivation behind self-nested LES (as also stated by the authors) is to resolve more turbulent scales. The finest grid resolution is chosen to be 1.25 m in the current study. What is this choice of grid resolution based on? As discussed by the authors on page 9 line 180, the chosen resolution does not fully resolve the inertial sub-range, which is understandable but then the question is what additional benefit could you get by partially resolving the inertial sub-range if you cannot reach the furthest smaller scales (like the dissipation scales), compared to only resolving the largest energy containing scales?

2. The authors choose a height of 119 m for comparison and call it as the 'hub height'. In practice, the hub height depends on a specific turbine model, and as the simulations are performed without a turbine, why is this height chosen for the comparison and referred to as the 'hub height'? As seen later in the manuscript, the turbulence quantities are compared at lower heights due to the measurement dataset, it is slightly confusing to see a particular height chosen as hub height, but then later not used for key comparisons.

3. The FINO1 data is available at the heights of 40, 60 and 80 m, and the authors extrapolate mean wind speed using logarithmic law and assume a constant variance. The accuracy of these approaches should be demonstrated in the article. Especially, how accurate is the assumption of a constant variance from 80 m upwards?

4. On page 6 line 130, the authors state that for onshore sites a power law exponent is 1/7. A reference to this figure should be provided, as a power law exponent can change much like the roughness length based on the land cover in onshore conditions.

5. On page 11 line 194-195, the authors hypothesize that the difference in the spectra between the measurements and simulations in the SBL case is due to the boundary layer height being below 119 m. As per my understanding, the comparison between simulated and measured spectra is done at 80 m, or is it that the measured spectra at 80 m are compared with simulated ones at 119 m? If so, how accurate is this comparison? Additionally, it is true that the SBL heights can be significantly lower than

the NBL and CBL ones, but are they actually below 80 m? A reference to back this should be provided, or this hypothesis should be revised.

6. On page 14 lines 223-224, the authors associate the difference between the simulated and measured power law exponents to the coarse measurement resolution. In order to back this up, the plots of simulated, simulation fitted power law, measured and measurement fitted power law should be shown (at least for a few simulations).

7. The last line of the conclusion (page 16 line 251) seems rather vague. What do the authors recommend? Either do only neutral simulations or do one-way nesting for stable and convective conditions? Do they recommend not using two-way nesting in any scenario?

8. A comparison of parameterized/unresolved turbulent scales could be added to understand how the unresolved scales change due to the nesting strategy.

**Minor Comments:**

Some minor technical comments are given here:

1. The abbreviations should be defined on first use. For example, the abbreviation of large-eddy simulation (LES) should be defined on page 1 line 16.
2. Figure 2 is not referenced in the text.
3. Probably missing 'child' in 'parent and domains' on page 8 line 167.
4. Probably missing 'with' between 'along other' on page 15 line 240.

---

## Author Response (AR2)

**Response to the comments**

Maria Krutova

May 2023

Thank you for the detailed reviews and valuable comments. We added clarifications in the parts that caused questions. Since the length of the simulation caused concerns, we also extended each case's main run to 4 hours of the simulated time so that the high-frequency time series covered period of $3-4\,\mathrm{h}$ instead of previously regarded $0-1\,\mathrm{h}$. In the main run, the flow development occurs in the first 10-20 minutes of the simulated time while the precursor flow passes the root domain once. All plots and values related to the main run are, therefore, recalculated and re-plotted. While we do not observe gradual changes in the spectral and coherence plots, flow evolution can be seen in the mean profiles – particularly, CBL profiles due to the influence from the turbulence.

**Reply to the reviewer comments 1 (RC1)**

**Major comments**

1. **The primary motivation behind self-nested LES (as also stated by the authors) is to resolve more turbulent scales. The finest grid resolution is chosen to be $1.25\,\mathrm{m}$ in the current study. What is this choice of grid resolution based on? As discussed by the authors on page 9 line 180, the chosen resolution does not fully resolve the inertial sub-range, which is understandable but then the question is what additional benefit could you get by partially resolving the inertial sub-range if you cannot reach the furthest smaller scales (like the dissipation scales), compared to only resolving the largest energy containing scales?**

   Initially, we aimed to reproduce the study by Nybø et al. (2020) in PALM. Nybø et al. (2020) was using SOWFA code for LES fields generation. We planned a cross-comparison between PALM, SOWFA and sonic anemometers data. Therefore, most of the domain parameters originate from the SOWFA study to keep the simulations comparable to each other and measurements. The issues with the two-way nesting under strong wind shear were interesting enough to report in a separate paper.

   While the two-way nesting effects can be noticed with only one child domain, defining $2-3$ child domains allows a more detailed comparison between one-way and two-way nesting modes. E.g., the spectra comparison shows that more scales can be resolved in the nested area of the coarsest domain if the two-way nesting is applied. At the same time, we also showed that refining the grid spacing below $5\,\mathrm{m}$ does not improve the accuracy enough to justify additional computational costs. Hence, further refinement is not justified for the free-flow areas, unless a specific grid spacing is required.

2. **The authors choose a height of 119 m for comparison and call it as the 'hub height'. In practice, the hub height depends on a specific turbine model, and as the simulations are performed without a turbine, why is this height chosen for the comparison and referred to as the 'hub height'? As seen later in the manuscript, the turbulence quantities are compared at lower heights due to the measurement dataset, it is slightly confusing to see a particular height chosen as hub height, but then later not used for key comparisons.**

   We chose the height of 119 m after Nybø et al. (2020) as an outlook into larger, 10 MW, wind turbines, particularly DTU 10MW reference wind turbine. Hence, all the measurement time series were selected by Nybø et al. (2020) to have similar free-flow wind speeds at the specific height, and we could re-use already processed time series. We agree that relying on these pre-defined characteristics may be confusing. For this study, we rename the 'hub height' of 119 m to the 'reference height' where the free-flow wind speeds are similar across the regarded cases.

3. **The FINO1 data is available at the heights of 40, 60 and 80 m, and the authors extrapolate mean wind speed using logarithmic law and assume a constant variance. The accuracy of these approaches should be demonstrated in the article. Especially, how accurate is the assumption of a constant variance from 80 m upwards?**

   After a discussion, we conclude that we cannot reliably estimate the turbulence intensity at 119 m with the available data. Since we are not performing a time series comparison for 119 m, we decided to remove the estimation of turbulence intensity value at 119 m. Section 2 now ends with a corrected paragraph:

   *The estimation is performed under an assumption of a boundary layer starting above 119 m and is applicable only to the mean wind profile. During the simulation, we attempt to match the mean wind profile, including the estimated wind speed at 119 m and turbulence intensity calculated for levels 40, 60, and 80 m.*

   The current LES setup produces turbulence intensity profiles within a 10% relative ($\sim$0.6% absolute value) deviation from the measurements, which we take as acceptable, considering that PALM does not allow specifying exact turbulence intensity via simple initialization.

4. **On page 6 line 130, the authors state that for onshore sites a power law exponent is 1/7. A reference to this figure should be provided, as a power law exponent can change much like the roughness length based on the land cover in onshore conditions.**

   This value was provided as an example to give a sense of the exponent's order. As was commented in the following line, that value is not a universal constant:

   *The power law exponent is sensitive to the atmospheric conditions and is usually approximated with the constants, e.g., $\alpha = 1/7$ for neutral onshore sites. Often, the approximations do not reflect seasonal and diurnal variations in the wind profiles (Bratton and Womeldorf, 2011; Jung and Schindler, 2021).*

   We add a reference to Touma (1977), which expresses the same concern on using 1/7 exponent and studies different stability conditions.

   *The power law exponent is sensitive to atmospheric conditions and is usually approximated with a constant, e.g., $\alpha = 1/7$ is applicable to neutral onshore sites but not other*

*stabilities (Touma, 1977). Often, the approximations do not reflect seasonal and diurnal variations in mean wind profiles (Bratton and Womeldorf, 2011; Jung and Schindler, 2021).*

5. **On page 11 line 194-195, the authors hypothesize that the difference in the spectra between the measurements and simulations in the SBL case is due to the boundary layer height being below 119 m. As per my understanding, the comparison between simulated and measured spectra is done at 80 m, or is it that the measured spectra at 80 m are compared with simulated ones at 119 m? If so, how accurate is this comparison?**

The comparison is performed for 80 m time series.

**Additionally, it is true that the SBL heights can be significantly lower than the NBL and CBL ones, but are they actually below 80 m? A reference to back this should be provided, or this hypothesis should be rev**

We reviewed the existing studies on low SBL and unnatural profiles, particularly, Muñoz-Esparza et al. (2012) and Kettle (2014), and examined 10-minute averaged profiles for the same period measured with the cup anemometer. The cup anemometer measurements reach up to 100 m, hence, we cannot define the SBL height exactly. After the review, we agree that the hypothesis should be retracted, as we do not see anomalies in the measured stable profiles below 100 m, which could be implying an unusual behavior of SBL.

[Figure]

Figure 1: Comparison of SBL precursor velocity profiles, original and reduced forcing run

We performed a test simulation of an SBL precursor for the same wind speed but weaker pressure gradient (-0.0001 Pa/m instead of -0.0005 Pa/m) and slightly stronger surface cooling (-0.3 K/s instead of -0.2 K/s). Due to a decreased forcing, the developed profiles deviated from the reference measurements and showed stronger shear but lower turbulence intensity (Fig. 1). However, we observed a better agreement with the measurements spectra (Fig. 2). Therefore, we are able to match only one of two: either SBL profiles or SBL spectra – and observe a strong discrepancy in another. We conclude that the cause of discrepancy could lie in the way PALM handles stable conditions.

To our knowledge, there are no throughout studies on SBL turbulence characteristics performed with up-to-date PALM code. Early PALM model (listed under code IMUK)

[Figure]

Figure 2: Comparison of SBL precursor spectra, original and reduced forcing run

participated in a cross-comparison of LES of SBL in Beare et al. (2006). While a comparison to observations was performed, it regarded only momentum, heat diffusivity, and fluxes. While Wurps et al. (2020) regards different grid resolutions and Ning et al. (2021) performs a comparison to SOWFA, both do not use measurements for verification and validation. Hence, simulating stable conditions in PALM with the respect to turbulence characteristics may require an additional verification study.

It should be noted that this conclusion refers only to a simple initialization via velocity and temperature profiles and surface cooling. Other ways of simulating SBL in PALM via dynamic driver from WRF data or synthetic turbulence behave differently.

Due to running extended simulations, we did not have enough CPU and real time left to perform an extensive comparison of the reduced forcing SBL simulation. We provide a brief comparison of a short non-nested SBL run in the Appendix of the revised article.

6. **On page 14 lines 223-224, the authors associate the difference between the simulated and measured power law exponents to the coarse measurement resolution. In order to back this up, the plots of simulated, simulation fitted power law, measured and measurement fitted power law should be shown (at least for a few simulations).**

   The line

   *The discrepancy between exact values of $\alpha$ in measurement and simulated fits could be explained by less precise power law fit in the measurement profiles: only three points were available for the fit.*

   was referring only to the low amount of levels in the sonics data. We compared fitting to sonics data with the cup anemometer data. While the latter has more levels available and thus may produce more accurate results, the resulting power law coefficients deviated farther from the LES value. A more plausible cause for the discrepancy lies in the definition of $U_{10}$ at $z = 10\,\text{m}$. This value is estimated for sonics data, while LES data can interpolate it from the available output. $U_{10}$ calculated from LES output deviates from the sonics estimation by $10-20\%$ and shifts the power law exponent.

7. **The last line of the conclusion (page 16 line 251) seems rather vague. What do the authors recommend? Either do only neutral simulations or do one-way**

**nesting for stable and convective conditions? Do they recommend not using two-way nesting in any scenario?**

The line

*We, therefore, recommend using either true neutral simulation or one-way nesting for the wind turbine wake simulation.*

is changed too

*We, therefore, recommend using the one-way nesting mode for the wind turbine wake simulation. In the case when the two-way nesting mode is preferable, only a true neutral setup does not produce secondary circulation.*

8. **A comparison of parameterized/unresolved turbulent scales could be added to understand how the unresolved scales change due to the nesting strategy.**

   Figures 3 and 5 present plots of resolved and subgrid scales. Subgrid-scale fluxes can be only output in PALM as spatially averaged profiles, hence the resolved-scale fluxes are also spatial averages from the same output.

   Nesting mode has a small effect on resolved fluxes in general. The effect is most pronounced in CBL simulations 3b, when comparing $\overline{wv}$ quantities.

   The subgrid-scale fluxes consistently remain near zero for all levels except few near-surface cells where the turbulence intensity remains high 5. Consequently, the near-surface subgrid-scale fluxes are comparable to resolved-scale fluxes. Yet, the subgrid-scale fluxes at lower levels tend to zero as the grid resolution is refined. The resolved fluxes in two-way nested run show non-monotonic behavior for the first point above the surface.

   **Plot update for an extended simulation and correction to the explanation**, Fig. 4: CBL two-way nested case shows strong evolution of the resolved-scale fluxes. While shorter simulations had similar behavior for both nesting modes, running LES for a longer time lead to a merging of two-way nested flux profiles, as should be expected in this mode.

   The SBL run also shows weaker evolution of the resolved fluxes, but only in the upper part of the domain. Lower levels covering heights of interest to the wind research are not affected.

   The near-surface fluxes (Fig. 5) are not strongly affected in the extended runs and retain a non-monotonic behavior in the lowest cells during two-way nested runs. In the initial response, this behavior was attributed to the spatial averages being affected by the local flow changes in the two-way nested area. However, the NBL run shows similar fluctuations in the near-surface resolved fluxes, although the flow is not altered in a true neutral simulation. Hence, non-smooth near-surface profile of the fluxes is likely caused by the anterpolation process, which is only carried on during the two-way nesting mode.

   The description of the resolved and subgrid-scale fluxes is added to the revised article as Section 4.2.

9. **New data upload and known issues**. The extended data sets are uploaded to `https://doi.org/10.5281/zenodo.7886678`. Because of technical issues, the two-way NBL simulation did not write a correct output for $\Delta t = 0.05\,\mathrm{s}$ for the last hour of the four-hour run. The plots in the revised article, therefore, use data from the previous hour, which

[Figure]

Figure 3: Comparison of resolved and subgrid-scale momentum fluxes for different stability simulations and nesting modes. One-hour simulations (old)

was output at a dynamic timestep of $\Delta t = 0.06 - 0.07\,\mathrm{s}$. The time series for spectra and coherence calculations are resampled at equal intervals. Despite the resampling, the graph shape for NBL two-way nesting is not affected in a noticeable way. The averaged cross-sections are recorded correctly.

The SBL two-way simulation had an $xz$ cross-section mistakenly defined outside of the nested area, so the profiles extracted for Figure 4 contradicted the values seen in an $xy$ cross-section. The revised article uses old data after one-hour simulation.

[Figure]

Figure 4: Comparison of resolved and subgrid-scale momentum fluxes for different stability simulations and nesting modes. Extended simulations (new)

[Figure]

Figure 5: Comparison of near-surface resolved and subgrid-scale momentum fluxes for different stability simulations and nesting modes

**Minor comments**

1. **The abbreviations should be defined on first use. For example, the abbreviation of large-eddy simulation (LES) should be defined on page 1 line 16.**

   Corrected

2. **Figure 2 is not referenced in the text.**

   The reference is added to line 159:

   *In the absence of the surface heat fluxes, i.e., in the true neutral case, the one-way and the two-way nested simulations behave similarly with the respect to grid spacing and feedback between domains (Fig. 2).*

3. **Probably missing 'child' in 'parent and domains' on page 8 line 167.**

4. **Probably missing 'with' between 'along other' on page 15 line 240.**

   The typos are corrected.

**Reply to the reviewer comments 2 (RC2)**

**Major comments**

1. **Line 40: "... at the hub height of 119 m" From the paper it does not get clear to me why 119 m are decribed as hub height. According to the information provided on `https://www.alpha-ventus.de/english` the hub heights in the wind farm alpha ventus that is situated next to the FINO1 met mast are 90 and 92 m, respectively. Moreover, I do not understand why not one of the real measurement heights is applied. Doesn't the introduction of the additional height to which measured data is extrapolated mean an additional source of uncertainty?**

   The estimated wind speed at 119 m serves to unify three different stability cases by the mean wind speed at the same level. No direct comparisons are performed at 119 m except of matching LES velocity profiles with the measurements. We detailed this purpose more strongly in the Data section and emphasized the available measurement levels in other sections.

   **Later in the paper it is written that the boundary layer height at the times of the measurements under stable stratification might be lower than 119 m. The equation in line 45 is however only applicable in the surface layer, i.e., in the lowest 10 percent of the boundary layer. From my point of view this arises doubts whether the derived 119 m data can actually be used for comparisons in the case of the stable boundary layer.**

   We do not possess measurements above 100 m for the regarded period of SBL conditions. Therefore, the actual height of SBL layer cannot be estimated reliably. The hypothesis on unusually low SBL was suggested to explain strong discrepancy between LES and measurement spectra. We performed a test simulation with matching SBL spectra but not profiles (Fig. 1 and 2). Currently, we are leaning to a conclusion, that defining SBL

conditions in PALM via initial parameters requires a verification study with the focus on turbulence characteristics.

2. **Figure 1: It seems as if the domain of the precursor run has exactly half the width of the domain of the main run in the neutral case. This setup is not ideal in order to break up long streaky structures. An excerpt from the PALM documentation reads: "Note that the initial flow field has a perfectly regular structure with a periodicity of the precursor run. This regularity can persist for a very long time. To break up this regularity, use a domain width that is not an integer multiple of the precursor run domain width. The occurring flow field discontinuity at the lateral domain boundary initiates a fast break up of the regularity." (`https://palm.muk.uni-hannover.de/trac/wiki/doc/app/examples/turbinf`). Figure 2 shows indeed elongated structures in the u- and the w-component that seem to extend over the whole length of the model domain. Are these structures realistic or only a result of the chosen setup? What does it mean for the comparability between simulations and measurements?**

The regularity can be broken by using parameter `y_shift` in the main run to ensure the breaking of the regular structures. This is not a precursor-specific parameter, and it is also recommended for the main run to ensure breaking of any possible regularity left in the precursor.

The structures forming in NBL and CBL runs are generally larger than in SBL due to higher turbulence intensity. Hence, the SBL run would look more uniform in the same color scale. The stripes forming in NBL and CBL averaged flows are not stationary and change over time.

3. **Line 74: Does the nesting procedure also apply linear interpolation of the wind speed components in the vertical direction close to the ground? Wouldn't it make more sense to apply a logarithmic interpolation here? A linear interpolation should actually result in an underestimation of the resulting wind speed, shouldn't it?**

Yes, the interpolation is linear and is performed similarly for all quantities and levels. The interpolation is performed only for the nested boundaries, consisting of the child domain's side and top surfaces. The bottom surface is always located at $z = 0$ m; PALM does not have an implementation of a free-hanging child domain. Hence, the bottom surface uses boundary conditions (Neumann or Dirichlet) prescribed in each child input parameter. Then the near-surface quantities are obtained when the prognostic equations are solved for a child domain.

Indeed, it would be interesting to check, whether introducing a logarithmic near-surface interpolation affects the nesting results. However, that requires modifying existing PALM code and for now falls outside the scope of initial study.

Points $4-7$ refer to the same concept of defining vertical profiles in PALM. We see these questions were raised due to an insufficient explanation of the precursor-main run approach in the article. The explanation was shortened initially to avoid focusing on the PALM

general concept that is not directly related to nesting. Seeing that it causes confusion, we expand Methodology section on the precursor.

The precursor-main run approach is chosen to ensure that nested and non-nested runs are initialized with exactly the same flow field. Simulating a precursor run in a smaller domain also saves computational time on generating steady turbulent inflow with required vertical profiles. Depending on the stability condition, the precursor may need to run for $2-6$ simulated days before the flow becomes stationary. The simulated flow is then used as an inflow for the main run, which does not have to be simulated for the same amount of time to study the nesting effects.

We add clarifications to Section 3.2 and expand Table 4 with the column 'Time' to reflect the simulation time required to reach a steady state in the precursor (Table 1). We also add another table to provide flow characteristics in the steady state, i.e., inflow of the main run: velocity at $119\,\mathrm{m}$, capping inversion, and surface temperature (Table 2).

Table 1: Input parameters of the precursor runs (Updated Table 4 from the original manuscript).

|  | $\overline{U}_0$, ms$^{-1}$ | $dp/dx$, Pa m$^{-1}$ | $z_0$, m | $T_s$, K | $\overline{w'\theta'}$, K ms$^{-1}$ | $dT_s/dt$, Ks$^{-1}$ | Run time, h |
|---|---|---|---|---|---|---|---|
| NBL (coarse) | 13.8 | $-2 \times 10^{-4}$ | $1.2 \times 10^{-3}$ | 300.0 | 0 | – | 27 |
| NBL (fine) | 14.1 | $-2 \times 10^{-4}$ | $1.7 \times 10^{-3}$ | 300.0 | 0 | – | 48 |
| CBL | 11.5 | $-1 \times 10^{-4}$ | $5 \times 10^{-4}$ | 281.3 | 0.015 | – | 146 |
| SBL | 13.0 | $-5 \times 10^{-4}$ | $8 \times 10^{-4}$ | 289.5 | – | -0.2 | 72 |

Table 2: Steady state of the precursor runs – turbulent inflow for the main run (New table).

|  | $\overline{U}_{119}$, ms$^{-1}$ | TI, % | $T_s$, K | $L$, m | Capping inversion, K/100 m |
|---|---|---|---|---|---|
| NBL (coarse) | 12.1 | 6.03 | 300 | $10^6$ | 0 |
| NBL (fine) | 12.9 | 6.94 | 300 | $10^6$ | 0 |
| CBL | 12.0 | 6.27 | 294.7 | -342 | 7.4 |
| SBL | 12.7 | 3.78 | 291 | 529 | 9 |

A detailed answer to each point is provided below.

4. **Line 101: How can there be a geostrophic wind without Coriolis force?**

   This is a collision of notations. The input parameters `ug_surface` and `vg_surface` are used in PALM to describe geostrophic wind near the surface and provide the initial state for the model. The initial profile is then considered a constant value along $z$-axis at $t = 0\,\mathrm{s}$. The initial profile changes due to the influence of external forces until the steady state is reached. When the Coriolis force is absent, the initial profile can still be prescribed with `ug_surface` and `vg_surface`. However, a pressure gradient should be added to compensate for the surface friction losses in the absence of the Coriolis force. The reason we chose the pressure gradient approach instead of the Coriolis force is explained in reply to point 7.

   To avoid confusion, the text in line 101

   *the geostrophic mean wind $\overline{U}$*

   is replaced by

   *the initial mean wind $\overline{U}_0$*

5. **Line 101: The simulations are run without a capping inversion. This is a difference to the situation in the real atmosphere. What does it mean for the comparability between the simulations and the measurements?**

   The prescribed temperature gradient serves only as an initial temperature profile. During the precursor run, the profile gets altered due to a heat flux (CBL case) or surface cooling (SBL case), resulting in a capping inversion. A true neutral simulation does not develop a capping inversion due to the temperature remaining constant during the simulation.

   We list a capping inversion in the new table describing the steady state in precursor runs used as the turbulence inflow for the main run (Table 2 in the response).

6. **Line 106: "We run main simulations for one hour" Is this sufficient to get a stationary solution? For that the inflow should not change with time and the flow should at least flow once through the model domain. I assume that the simulation time used is actually to short. What was the averaging period used? From my point of view it does not make sense to start with the averaging period directly at the beginning of the large-eddy simulation.**

   We use the precursor-main run approach. The long precursor run generates a steady flow field with the required wind speed and turbulence profiles. In this run, the vertical profiles are affected by the pressure gradient, surface friction, and heat flux.

   The main run copies the averaged stationary profiles from the last step of the precursor run (assumed to reach a steady state) to initialize the flow field in a larger domain. The turbulent inflow is then recycled over the main run domain. The averaged profiles generated by the precursor remain constant with random fluctuations added to them (Fig. 6). The fluctuations are rather strong in NBL and CBL runs due to naturally stronger turbulence. Overall, the instantaneous velocity profiles do not deviate far from the original inflow profile. The fluctuations in the temperature profile remains negligible in the main run.

   We simulated an extended CBL two-way nested run, but did not observe any development in the inertial subrange of the spectra. The fluctuations in the energy containing eddies are explained by the length of the time series – one hour to match the measurement time series. Indeed, Hellsten et al. (2021) ran their nested simulations for longer time. Yet, they either studied smaller velocities, i.e., require more time for the flow to pass the domain ones, or were interested in the flow development (cubes case, flow over hill). Our interest is primarily turbulence statistics which were less affected by the extended simulations.

   It was, however, an oversight on our part, as there is definitely a quick development happening in the beginning of the main run. To obtain one hour time series for the comparison, the original runs should have been extended by at least half-hour.

   Since the time for revision allows this, we extended the existing simulation to four hours to get new time series without any interference from the domain initialization.

7. **Line 173-176: It did not get clear to me why the Coriolis force had been omitted in the simulations. Moreover, I miss a discussion on how strong the impact of neglecting the Coriolis force actually is and what it means for the comparability of the measurements and the simulations. I recommend to add a thorough discussion of the topic to the manuscript.**

[Figure]

Figure 6: Example profiles from CBL two-way nested run, root domain. The intermediate profiles are plotted with a step of $60\,\mathrm{s}$ in the period $0.5 - 1.5\,\mathrm{h}$ from the start of the main run. The lower row shows the same profiles zoomed into the surface layer. Averaged velocity profile is provided for the comparison.

The measurements time series were already rotated, so that the $x$-axis aligned with the series average wind direction.

We needed to closely match the velocity and turbulence profiles of the measurement data. PALM does not possess the capability to define turbulence intensity profiles in a simple initialization. Instead, the profiles develop naturally during the simulation based on the input parameters. Several re-runs may be required to tune the parameters (initial velocity, pressure gradient, heat flux, and roughness length) until the desired profiles are reached in the steady state. Using the Coriolis force also requires accounting for the flow rotation and adjusting wind speed components to match the wind direction in the measurements.

A non-neutral precursor run takes a considerable amount of time, e.g., 10 hours of real time, to simulate $\sim$6 days until the steady state of CBL. Depending on the initial guess, the precursor may require several attempts until the simulation produces velocity and turbulence profiles close to the measurement data. Considering this, we preferred a simpler approach by replacing the Coriolis force with the pressure gradient along the $x$-axis. This way, we do not have to consider the flow rotation, i.e., reduce the number of input

[Figure]

Figure 7: Example CBL spectra from 4-hour two-way nested simulation.

parameters to be tuned. The main outcome of this decision are weaker fluctuations of $v$-components compared to the measurement series. We attempted to compensate for this by considering total horizontal velocity instead of regarding $u$ and $v$-components separately in the measurement series.

**Minor comments**

1. **Language check: Please check carefully the use of articles again. From my point of view in the current version of the manuscript the article "the" is used in many places where it would usually not be used.**

   The manuscript is checked again for the revised submission, various articles usage and wording are corrected.

2. **According to the information provided in the manuscript a deviation between**

the mean wind speed in the measurements and in the simulations is obvious (see table 1 and table 4). How does this impact the meaningfulness of the results? It would be good to add a discussion on that topic. At least it should be stated that even the mean flow condition observed in the experimental data is not met by the mean flow in the simulations.

The input parameters passed to PALM do not define the steady flow but only serve to set the initial profile. Due to the surface friction, pressure gradient, and (if present) heat fluxes, the profile is altered over time in the precursor run. We simulate the precursor run until the velocity components reach a steady state.

To avoid confusion, we rename the column $\overline{U}, \mathrm{ms}^{-1}$ in Table 4 (see Table 1 in this response) to $\overline{U}_0, \mathrm{ms}^{-1}$ in order to show that this is not the final value. We also add value of $\overline{U}_{119}$ to provide the steady state velocity at $119\,\mathrm{m}$ (see an additional Table 2).

3. **Line 31: Please change "... are then compared between the model results and measurements ..." to "... are then compared with measurements"**

   The wording is corrected.

4. **Line 60: "Universität Hanover" $->$ Please change either to "Universität Hannover" (my suggestion) or "Hanover University" or "University of Hanover".**

   The typo is corrected to *Universität Hannover*

5. **Line 78: "In the two-way nesting case, each child domain interpolates its solution back to the respective parent domain" Wouldn't it make sense to include also an averaging process in bringing the fine-grid data to the coarse grid? It should be considered that PALM uses the volume-averaging approach.**

   PALM model description and documentation use a term 'anterpolation' for the process of returning a child's domain solution to its parent. Here, the anterpolation means first-order numerical integration of the child cells within each parent cell. Since it is not a common term in LES, we avoided using it, which lead to oversimplification and an erroneous description of the process. The corresponding paragraph

   *In the two-way nesting case, each child domain interpolates its solution back to the respective parent domain before the pressure-correction step. Therefore, the two-way nested solution remains similar in the nested area, while the one-way nested solution may eventually diverge for parent and child domains.*

   is corrected to

   *In a two-way nesting case, PALM uses an anterpolation scheme proposed by Clark and Farley (1984) to return a child domain's solution to the parent domain; technical details behind the implementation are explained in Hellsten et al. (2021). Each child domain anterpolates its solution via first-order integration to the respective parent domain before the pressure-correction step. Therefore, the two-way nested solutions remains similar in the nested area, while the one-way nested solution may eventually diverge for parent and child domains.*

6. **Line 83: "development and" $->$ "development"**

7. **Line 84: "reaches steady state" −> "reaches a steady state"**

   The typos are corrected

8. **Line 103: The SBL case uses surface cooling over time. Does it mean that the inflow changes with time? If so, wouldn't this create vertical movements? In case that indeed vertical movements are observed, are any measures taken to damp those movements?**

   The temperature changes only during the precursor phase. The turbulent inflow in the main run is not affected and is only subjected to random fluctuations.

9. **Line 111: I suggest using the notation $a(t, x1)$ $a(t, x2)$ instead of $a(t)$ and $b(t)$. $a$ and $b$ are introduced once as time series and once as points. The points however do not change with time.**

   Since the specific locations are not mentioned (only the distance between points is important and not many levels are available for the comparison), and all expressions are presented with a focus on the time series, we correct the wording

   *points a and b*

   to

   *time series $a(t)$ and $b(t)$*

10. **Line 117: Please change "synchronity" to "synchronicity".**

    The typo is corrected.

11. **Table 5: Please add an information on the grid spacing used in the CBL and in the SBL cases.**

    The table (new index – Table 6 due to an addition of a new table) is re-formatted to add $\Delta_x$ column.

12. **Figure 3: It seems to me as even in the case of the one-way nesting the flow field is developing between the inflow boundary and the outflow boundary. The values of the u-component close to the outflow boundary seem to be lower than the values close to the inflow boundary. Is this behavior expected?**

    This effect also appears in the non-nested simulation. The decrease is relatively small ($\sim 0.5\,\mathrm{m/s}$ or 5% of the mean wind speed). We consulted PALM developers on this question – the effect lies within expectations for the PALM version used. The slow down can be controlled by altering the pressure gradient in the main run. However, the size of the main run's domain and required CPU time for the tests make this adjustment impractical. Therefore, we leave the original parameters.

    Since a weak slowing effect along the domain emerges in all simulation modes (non-nested, one-way or two-way nesting), it is not a direct cause of the slow down observed in a two-way simulation and we can still assume that the free flow in the root domain behaves similarly within the same stability case.

13. **Line 233: "We performed nested LES of three stability cases for the same wind mean wind speed of 12.5 ms-1". I do not see this statement be supported by**

**what is reported in the paper. Neither the measurements nor the simulations show a mean wind speed of 12.5 ms-1. Please revise the statement.**

We add Table 2 to show the parameters of the turbulence inflow and their difference from the precursor input parameters in Table 1. The wind speed at $119\,m$ is not exactly $12.5\,\mathrm{ms^{-1}}$, but the variation between cases is lower than could be expected from Table 1, which lists precursor parameters.

We also change the line from *We performed nested LES of three stability cases for the same wind mean wind speed of* $12.5\,\mathrm{ms^{-1}}$ to *We performed nested LES of three stability cases for the horizontal mean wind speed of* $12-13\,\mathrm{ms^{-1}}$ *at the reference height 119 m*

14. **Line 256 and 260: "Deutsches Windenergi Institut" −> "Deutsches Windenergie Institut"**

    The typo is corrected

**Additional revisions**

While revising the article, we found several paragraphs that required correction.

1. **SBL domain**. The domain for SBL simulation was extended to review the slowing of the free flow observed in nested and non-nested cases, as mentioned by the second reviewer. With the extended domain, we were able to confirm that the slowing is persistent and is not an occasional fluctuation of the averaged flow. We decided to replace the old SBL domain (Fig. 8) with the extended one for new simulations (Fig. 9). The length of the new domain allows resolving fully a speed-up zone behind the nested area. The domain characteristics and required computational time were updated accordingly.

[Figure]

Figure 8: Nested domains schematic. (a) NBL and CBL domains, (b) SBL domains (old).

2. **Precursor parameters** (Section 3.2). Besides an additional table on the precursor steady state, a line about initial temperature gradients was added to Section 3.2.

3. **Notation correction** (Section 3.3). Because of an oversight, the coherence and co-coherence were marked with the same symbol in Eq.(4)−(6). This is now corrected to avoid confusion.

[Figure]

Figure 9: Nested domains schematic. (a) NBL and CBL domains, (b) SBL domains (new). The updated size allows plotting in the same scale.

*NBL and SBL cases start with zero temperature gradient; CBL case has initial temperature gradient of $1\,\mathrm{K}/100\mathrm{m}$.*

4. **CPU time** (Section 4.1). The CPU time per second of the simulated time is provided for simulations run at a fixed time step of $\Delta t = 0.05\,\mathrm{s}$. A paragraph is added to remark the CPU-time comparison is performed only for high-frequency output.

*It should be noted that, unless obtaining high-frequency time series is the main goal of a simulation, the time step can be gradually increased for non-nested runs in order to speed up the computation. The computational time will, nevertheless, increase in a similar proportion with the global grid refinement. The time step in nested runs is still limited by the lowest grid spacing in child domains. E.g., the dynamic step in the regarded configuration will not exceed $0.075\,\mathrm{s}$ to satisfy Courant–Friedrichs–Lewy condition.*

5. **True neutral conditions and two-way nesting mode** (Section 4.1). A remark is added to note that the simulation conditions have to be explicitly set to the true neutral for the two-way nesting affects to be absent. Simulating neutral conditions without the true neutral flag results in a flow slowing in the two-way nested area similar to observed in CBL and SBL conditions, even if the heat fluxes are absent and the temperature gradient is set zero.

Old paragraph

*Depending on the domain configuration, LES produces different results in the area of the refined grid. In the absence of the surface heat fluxes, i.e., in the true neutral case, the one-way and the two-way nested simulations behave similarly with the respect to grid spacing and feedback between domains. When the heat fluxes are introduced for the CBL and SBL simulations, the two-way simulation results in the decreased flow speed in the child domains.*

New paragraph

*Depending on the simulation conditions, LES produces different results in the nested area. If the true neutral case is defined in PALM explicitly via setting a corresponding flag, the*

*one-way and the two-way nested simulations behave similarly with respect to grid spacing and feedback between domains. Switching on the true neutral flag means that the temperature equation and buoyancy terms are not considered in the calculations. As long as those terms are introduced for non-neutral simulations, the two-way nested simulation results in a decreased flow speed in child domains.*

6. **Anisotropy (Section 3.4, Section 4.3.2)**. The anisotropy estimation was introduced in Section 3.4 as:

   *We estimate turbulence anisotropy by comparing spectra of the velocity components for the reduced frequency $f_r > 1$. Since the LES spectra does not resolve the inertial subrange fully, we take the bin-averaged spectra and select the bin at the beginning of the range $fz/U_z > 1$ for $z = 80$ m. We compute ratios $S_{vv}/S_{uu}$ and $S_{ww}/S_{uu}$ for all regarded cases.*

   This paragraph was corrected to:

   *We estimate turbulence anisotropy by comparing spectra of velocity components for the normalized frequency $f_n = fz/U_z$, where $z = 80$ m and $U_z$ is the horizontal velocity at this level. We compute ratios $S_{vv}/S_{uu}$ and $S_{ww}/S_{uu}$ for all regarded cases at $f_n \approx 1$.*

   The frequency was changed to 'normalized frequency' $f_r$ to avoid confusion with the reduced frequency $f_r$ which, being calculated similarly for the coherence plots, uses the vertical separation distance $\delta$ instead of the level height $z$.

   The new paragraph emphasizes calculation at the point $f_n \approx 1$, not a range $f_n > 1$. In addition, the spectra used for calculation were changed from bin-averaged spectra to Welch spectra. The bin-averaged and Welch spectrum agree well in the resolved inertial subrange, but interpret low and high frequencies differently (Fig. 10a). The bin-averaged spectra preserves low frequencies but returns fewer points in the frequency range making the selection of a point $f_n \approx 1$ less precise.

   The corrected estimation mostly preserves trends calculated from bin-averaged spectra (Fig. 10bc), but gives a clearer distinction between one-way and two-way cases: two-way cases show lower variation along the column as should be expected from flows remaining similar. The anisotropy for new time series (Fig. 10d) amplifies the differences between one-way and two-way nesting modes.

   Section 4.3.2 is updated accordingly to describe the new table.

[Figure]

(a) $u$-spectrum, non-nested NBL, $\Delta = 10\,\text{m}$. Marked is the frequency $f$ matching $f_n = fz/U_z = 1$

(b) Original estimation from bin-averaged spectra, old data – $0-3600\,\text{s}$ series

(c) Corrected estimation from Welch spectra, old data – $0-3600\,\text{s}$ series

[Figure]

(d) Corrected estimation from Welch spectra, new data – $10\,800-14\,400\,\text{s}$ series

Figure 10: Anisotropy estimation from the power density spectra

**Corrections to final revision**

These corrections were implemented into the accepted manuscript 31.05.2023.

**Requested corrections**

**page 3 lines 49-55 need clarity. The authors define a reference height 119 m for U_119. But then for log fitting they define another reference height at 80 m. Using similar terms can be a little confusing. In addition, the statement that boundary layer starts at 119 m is a little confusing. Of course, there will be boundary layer below 119 m, so what exactly is meant by this statement?**

The parameters $u_{ref}$ and $z_{ref}$ were renamed into $u_{F1}$ and $z_{F1}$ so that the corresponding part

*…the mean wind speed profile was approximated by Nybø et al. (2020) by fitting the logarith-*

*mic law*

$$\overline{u}(z) = u_{ref} \left[ \frac{\ln\left(\dfrac{z}{z_0} - \psi\right)}{\ln\left(\dfrac{z_{ref}}{z_0} - \psi\right)} \right] \tag{1}$$

*where the reference wind speed $u_{ref}$ is taken for the reference height $z_{ref} = 80\,\mathrm{m}...$*

was changed into

*...the mean wind speed profile was approximated by Nybø et al. (2020) by fitting the logarithmic law*

$$\overline{u}(z) = u_{F1} \left[ \frac{\ln\left(\dfrac{z}{z_0} - \psi\right)}{\ln\left(\dfrac{z_{F1}}{z_0} - \psi\right)} \right] \tag{2}$$

*where the wind speed $u_{F1}$ measured at FINO1 is taken for the highest available level $z_{F1} = 80\,\mathrm{m}...$*

Following the second part of the comment, the line

*The estimation is performed under an assumption of a boundary layer starting above $119\,m$ and is applicable only to the mean wind profile.*

was changed into

*The estimation is performed under an assumption of a boundary layer starting above $119\,m$ and is applicable only to the mean wind profile.*

**Some additional technical corrections for the references:**
**- page 20, line 316: "danish" $->$ "Danish"**
**- page 21, line 355: spurious "https://doi.org/"**
**- page 21, line 359: spurious "https://doi.org/"**
Corrected. The errors were caused by incorrectly filled fields in the bibliography file.

**Additional correction**

It appeared, that NBL and SBL profiles showing momentum fluxes were plotted from an intermediate output, thus not matching the time step of CBL profiles $t = 14\,400\,\mathrm{s}$ (Fig. 11, 13). Since the change in profiles is subtle and does not affect the conclusions, Fig. 5−6 were re-plotted for consistency (Fig. 12, 14). Old figures are also provided in the corresponding archive as `fig005_old.eps` and `fig006_old.eps`. However, it should be noted that the plotting scripts and output provided will reproduce the new figures when run.

The comparisons are provided in the next pages.

[Figure]

Figure 11: **OLD FIGURE** Comparison of resolved and subgrid-scale momentum fluxes for different stability simulations and nesting modes

[Figure]

Figure 12: **NEW FIGURE** Comparison of resolved and subgrid-scale momentum fluxes for different stability simulations and nesting modes

[Figure]

Figure 13: **OLD FIGURE** Comparison of near-surface resolved and subgrid-scale momentum fluxes for different stability simulations and nesting modes

[Figure]

Figure 14: **NEW FIGURE** Comparison of near-surface resolved and subgrid-scale momentum fluxes for different stability simulations and nesting modes

**References**

R. J. Beare, M. K. Macvean, A. A. Holtslag, J. Cuxart, I. Esau, J. C. Golaz, M. A. Jimenez, M. Khairoutdinov, B. Kosovic, D. Lewellen, T. S. Lund, J. K. Lundquist, A. McCabe, A. F. Moene, Y. Noh, S. Raasch, and P. Sullivan. An Intercomparison of Large-Eddy Simulations of the Stable Boundary Layer. *Boundary-Layer Meteorol. 2006 1182*, 118(2):247–272, 5 2006. ISSN 1573-1472. doi: 10.1007/S10546-004-2820-6.

D. C. Bratton and C. A. Womeldorf. The wind shear exponent: Comparing measured against simulated values and analyzing the phenomena that affect the wind shear. In *ASME 2011 5th Int. Conf. Energy Sustain. ES 2011*, pages 2245–2251. American Society of Mechanical Engineers Digital Collection, 3 2011. ISBN 9780791854686. doi: 10.1115/ES2011-54823.

T. Clark and R. Farley. Severe downslope windstorm calculations in two and three spatial dimensions using anelastic interactive grid nesting: A possible mechanism for gustiness. *J. Atmos. Sci.*, 41:329–350, 1984.

A. Hellsten, K. Ketelsen, M. Sühring, M. Auvinen, B. Maronga, C. Knigge, F. Barmpas, G. Tsegas, N. Moussiopoulos, and S. Raasch. A nested multi-scale system implemented in the large-eddy simulation model PALM model system 6.0. *Geosci. Model Dev.*, 14(6): 3185–3214, 6 2021. ISSN 1991-9603. doi: 10.5194/gmd-14-3185-2021.

C. Jung and D. Schindler. The role of the power law exponent in wind energy assessment: A global analysis. *Int. J. Energy Res.*, 45(6):8484–8496, 5 2021. ISSN 1099-114X. doi: 10.1002/ER.6382.

A. J. Kettle. Unexpected vertical wind speed profiles in the boundary layer over the southern North Sea. *J. Wind Eng. Ind. Aerodyn.*, 134:149–162, 2014. ISSN 01676105. doi: 10.1016/j. jweia.2014.07.012.

D. Muñoz-Esparza, B. Cañadillas, T. Neumann, and J. van Beeck. Turbulent fluxes, stability and shear in the offshore environment: Mesoscale modelling and field observations at FINO1. *J. Renew. Sustain. Energy*, 4(6):063136, 11 2012. ISSN 1941-7012. doi: 10.1063/1.4769201.

X. Ning, M. Krutova, and M. Bakhoday-Paskyabi. Analysis of offshore wind spectra and coherence under neutral stability condition using the two LES models PALM and SOWFA. In *J. Phys. Conf. Ser.*, volume 2018, page 012027. IOP Publishing, 9 2021. doi: 10.1088/ 1742-6596/2018/1/012027.

A. Nybø, F. G. Nielsen, J. Reuder, M. J. Churchfield, and M. Godvik. Evaluation of different wind fields for the investigation of the dynamic response of offshore wind turbines. *Wind Energy*, 23(9):1810–1830, 7 2020. ISSN 10991824. doi: 10.1002/we.2518.

J. S. Touma. Dependence of the wind profile power law on stability for various locations. *J. Air Pollut. Control Assoc.*, 27(9):863–866, 7 1977. ISSN 00022470. doi: 10.1080/00022470. 1977.10470503.

H. Wurps, G. Steinfeld, and S. Heinz. Grid-Resolution Requirements for Large-Eddy Simulations of the Atmospheric Boundary Layer. *Boundary-Layer Meteorol.*, 175(2):179–201, 5 2020. ISSN 15731472. doi: 10.1007/s10546-020-00504-1.